# Cohort studies on 71 outcomes among people with atopic eczema in UK primary care data

Julian Matthewman [1] ✉, Anna Schultze [1], Helen Strongman [1], Krishnan Bhaskaran [1], Amanda Roberts[2], Spiros Denaxas [3,4,5], Kathryn E. Mansfield [1] & Sinéad M. Langan [1]

Atopic eczema may be related to multiple subsequent adverse health outcomes. Here, we provide evidence to judge and compare associations between eczema and a comprehensive set of outcomes. We conducted 71 cohort studies (age, sex, general practice-matched) using Clinical Practice Research Datalink Aurum primary care records (1997–2023), comparing up to 3.6 million people with eczema to 16.8 million without. Eczema was associated with subsequent diagnosis of outcomes with adjusted hazard ratios (99% confidence intervals) from Cox regression of up to 4.02(3.95–4.10) for food allergy (rate difference [RD] per 1,000 person-years of 1.5). Besides strong associations with atopic and allergic conditions (e.g., asthma 1.87[1.39–1.82], RD5.4) and skin infections (e.g., molluscum contagiosum 1.81[1.64–1.96], RD1.8), the strongest associations were with Hodgkin's lymphoma (1.85[1.66–2.06], RD0.02), Alopecia Areata (1.77[1.71–1.83], RD0.2), Crohn's disease (1.62[1.54–1.69], RD0.1), Urticaria (1.58[1.57–1.60], RD1.9), Coeliac disease (1.42[1.37–1.47], RD0.1), Ulcerative colitis (1.40[1.34–1.46], RD0.1), Auto-immune liver disease (1.32[1.21–1.43], RD0.01), and Irritable bowel syndrome (1.31[1.29–1.32], RD0.7). Sensitivity analyses revealed the impact of consultation bias or choice of cohort age cut-off on findings. Comparatively large HRs in severe eczema were seen for some liver, gastrointestinal and cardiovascular conditions, osteoporosis, and fractures. Most cancers and neurological conditions were not associated with eczema.

Eczema, also referred to as atopic eczema or atopic dermatitis, is one of the most common chronic conditions worldwide[1], and is associated with a substantial morbidity burden and cost for health care systems[2]. Eczema, besides being associated with atopic diseases such as allergies and asthma, may also be associated with non-atopic diseases, possibly due to mechanisms such as chronic inflammation (which could explain observed cardiovascular outcomes)[3], psychological stress, low self-esteem, and sleep deprivation (which could explain observed anxiety and depression outcomes)[1,4]. Guidelines published by the American Academy of Dermatology (AAD) in June 2022 included statements on 32 different adverse health outcomes, for each judging whether an association is likely to exist and the quality of the evidence[5]. While there was clear evidence for associations between eczema and other atopic conditions (e.g., asthma and food allergies) the prior evidence for most adverse health outcomes included in the review (including mental illness, cardiovascular disease, metabolic disease, osteoporosis and fractures, and

[1]London School of Hygiene & Tropical Medicine, London, UK. [2]Independent Patient Partner, London, UK. [3]Institute of Health Informatics, University College London, London, UK. [4]NIHR UCLH BRC, London, UK. [5]BHF Data Science Centre, HDR UK, London, UK. ✉e-mail: julian.matthewman@lshtm.ac.uk

skin infections) was less clear. A hypothesis-generating study published in the same year suggests associations with gastrointestinal and neurological conditions which weren't included in the guidelines[6], and other important outcomes may exist but may have not been discovered. There is no internationally accepted approach to screening and prevention of adverse outcomes[1], despite potentially substantial impact at reducing morbidity and costs for health care systems, given eczema is common.

Studies on a range of health outcomes linked to eczema have typically focused on single, or small sets of, outcomes. Here, using the Clinical Research Practice Datalink (CPRD) Aurum, we employed best-practice epidemiological study design, an outcome-wide confounding-adjustment strategy, and suitable approaches to sensitivity and secondary analyses across 71 outcomes to efficiently and systematically generate high-quality evidence on associations.

## Results

### Descriptive Statistics

From the CPRD Aurum population ($N = 46,795,888$), we identified 3,823,770 individuals meeting the eczema definition (at least one record of an eczema diagnostic code and at least two records for eczema therapies) who were eligible for matching, and were matched with eligible unexposed individuals, resulting in a cohort of 20,398,809 (with and without eczema) for the any-age cohort (Fig. 1). Individuals were followed up for a median (IQR) of 4.7 (1.8, 9.9) years per person in the any-age cohort, 4.3 (1.7, 9.1) for the 18+ cohort (i.e., including only individuals aged 18 or older), 5.7 (2.4, 10.7) for the 40+ cohort (i.e., including only individuals aged 40 or older) (see histograms in Supplementary Fig. 3 [age at index date], and Supplementary Fig. 4 [follow-up time]). After matching, cohorts were balanced in terms of age and sex, but there were differences in comorbidities (e.g., previous asthma 8.6% in unexposed versus 17% in exposed) (Table 1).

### Associations between eczema and adverse health outcomes

For all outcomes, comorbidity-adjusted (i.e., adjusted for history of each other outcome at at index date) hazard ratios from Cox regression (i.e., the relative increase in hazard in the exposed) with 99% confidence intervals, and estimated rate differences (RD) (i.e., the number of additional outcomes experienced by the exposed) per 1,000 person-years from their respective main cohorts are shown in Fig. 2, Supplementary Table 1, and described by category in Supplementary Notes 1. Associations were strongest for food allergy (adjusted HR [aHR] 4.02, 99% confidence-interval [3.95–4.10]), allergic conjunctivitis (2.02 [1.99–2.05]), and for allergic rhinitis (1.93 [1.91-1.94]). Outcomes with hazard ratios closest to the null included prostate cancer (aHR 1.01 [0.99–1.04]), breast cancer (aHR 1.03 [1.01–1.06]) and Parkinson's disease (aHR 1.02 [0.98-1.06]). Estimated rate differences based on the adjusted hazard ratio were highest for allergic rhinitis (5.4 per 1,000 person-years), asthma (5.4) and dermatophyte infections (3.8). Comorbidity-adjusted hazard ratios were generally attenuated as compared to minimally-adjusted hazard ratios (Fig. 2, Supplementary Table 1).

Outcomes that were not included in the 2022 AAD guidelines (on comorbidities associated with eczema)[5] where we found all three of 1. strong confounder-adjusted associations (aHR > 1.2), 2. dose-response relationships (with eczema severity), and 3. considerable absolute rate differences (RD ≥ 0.48) were: irritable bowel syndrome (aHR 1.31 [1.29–1.32]; RD 0.67), oesophagitis (aHR 1.25 [1.23–1.27]; RD 0.48), gastro oesophageal reflux disease (aHR 1.25 [1.24–1.26]; RD 1.10), thromboembolic disease (aHR 1.25 [1.23–1.27]; RD 0.51), obesity (aHR 1.22 [1.21–1.23]; RD 0.77), chronic obstructive pulmonary disease (aHR 1.22 [1.20–1.23]; RD 1.15), gastritis and duodenitis (aHR 1.21 [1.20–1.23]; RD 0.60) and peripheral neuropathies (aHR 1.21 [1.20–1.22]; RD 2.13). Associations with larger hazard ratios (aHR > 1.3), dose-response relationships, with lower rate differences (RD ≤ 0.10) included: Hodgkin's lymphoma (aHR 1.85 [1.66–2.06]; RD 0.02), Crohn's disease (aHR 1.62 [1.54–1.69]; RD 0.09), coeliac disease (aHR 1.42 [1.37–1.47];

RD 0.10), ulcerative colitis (aHR 1.40 [1.34–1.47]; RD 0.08), and auto-immune liver disease (aHR 1.32 [1.21–1.43]; RD 0.01).

Results from sensitivity analyses that used the other cohorts and models additionally adjusted for drugs (history of oral glucocorticoid and systemic immunosuppressant prescriptions at index date) are shown in Fig. 3, Supplementary Tables 2 and 3, and described by category in Supplementary Notes 1. Models additionally adjusted for drugs did not considerably change results for any outcomes. Results from the any-age, 18+ and 40+ cohorts were similar for most outcomes, however for some outcomes there were considerable changes, e.g., food allergy (any-age cohort 4.02[3.95–4.10]; 18+ cohort: 2.03 [1.96–2.10]; 40+ cohort: 1.66 [1.58–1.74]). Results from the more severe cohort (i.e., individuals were considered exposed when they had a record indicating more severe eczema after meeting the eczema definition) were generally similar to results from the respective main analysis. Results from the <18 cohort (i.e., a subset of the any-age cohort of individuals that met the eczema definition before their 18th birthday and their matched comparators) varied; for atopic and allergic outcomes HRs were larger, while for several other outcomes HRs were attenuated as compared to their respective main analysis. When non-consulters (i.e., individuals who did not have a record indicating a primary care consultation in the year before index date) were excluded, HRs were attenuated across most outcomes.

### Associations between mild, moderate, and severe eczema and adverse health outcomes

Results from the secondary analysis of time-updated eczema severity are shown in Fig. 3, Supplementary Table 4, and described by category in Supplementary Notes 1. Outcomes which were found to be strongly associated with eczema (e.g., food allergy) were generally found to be more strongly associated with moderate (additionally hospital-admission-adjusted aHR 4.19 [4.06–4.32]), and severe eczema (5.72 [4.81–6.80]) than mild eczema (3.66 [3.59–3.75]). For some less strongly associated outcomes, results did not suggest a dose-response relationship with more severe eczema (e.g., migraine: mild 1.12 [1.11–1.14]; moderate 1.13 [1.11–1.15]; severe 1.03 [0.95–1.13]), while for others they did (e.g., peripheral artery disease: mild 1.06 [1.03–1.10]; moderate 1.12 [1.09–1.15]; severe 1.54 [1.36–1.75]). For some outcomes, including some that were strongly associated, confidence intervals for severe eczema were wide (Hodgkin lymphoma: mild 1.13 [0.95–1.35]; moderate 1.58 [1.37–1.82]; severe 2.02 [1.03–3.98]).

### Benchmarking against previous studies

Adjusted hazard ratios from our study were very similar to those from previous studies that used the similar CPRD GOLD database with similar study designs, but bespoke covariate selection. The CIs from our study were almost all within the CIs from the CPRD GOLD studies (Supplementary Fig. 1). We compared with studies on (1) anxiety (our aHR 1.16 [1.16–1.17], their aHR 1.17[1.14–1.19]) and Depression (our aHR 1.16 [1.15–1.17], their aHR 1.14[1.12–1.16])[4]; (2) cardiovascular outcomes, including myocardial infarction (our aHR 1.09 [1.07–1.11], their aHR 1.06[0.98–1.15]), heart failure (our aHR 1.17 [1.15–1.19], their aHR 1.19[1.10–1.30]) and stroke (our aHR 1.09 [1.08–1.11], their aHR 1.10[1.02–1.19])[3]; (3) fracture outcomes (e.g., Hip fracture: our aHR 1.10 [1.08–1.13], their aHR 1.09[1.06–1.12])[7]; (4) and cancer outcomes, where there was also no association with solid organ cancers (e.g., lung cancer: our aHR 1.05 [1.02–1.08], their aHR 1.08[1.01–1.16]; breast cancer: our aHR 1.03 [1.01–1.06], their aHR 0.99[0.94–1.04]; prostate cancer: our aHR 1.01 [0.99–1.04], their aHR 1.06[1.00–1.13]), but associations with non-melanoma skin cancer (our aHR 1.14 [1.12–1.15], their aHR 1.11[1.06–1.15]) and Non-Hodgkin lymphoma (our aHR 1.26 [1.21–1.32], their aHR 1.20[1.07–1.34]) and a strong association with Hodgkin lymphoma (our aHR 1.83 [1.64–2.04], their aHR 1.48[1.07–2.04]), however with wider confidence intervals than in our study[8]. We also found that

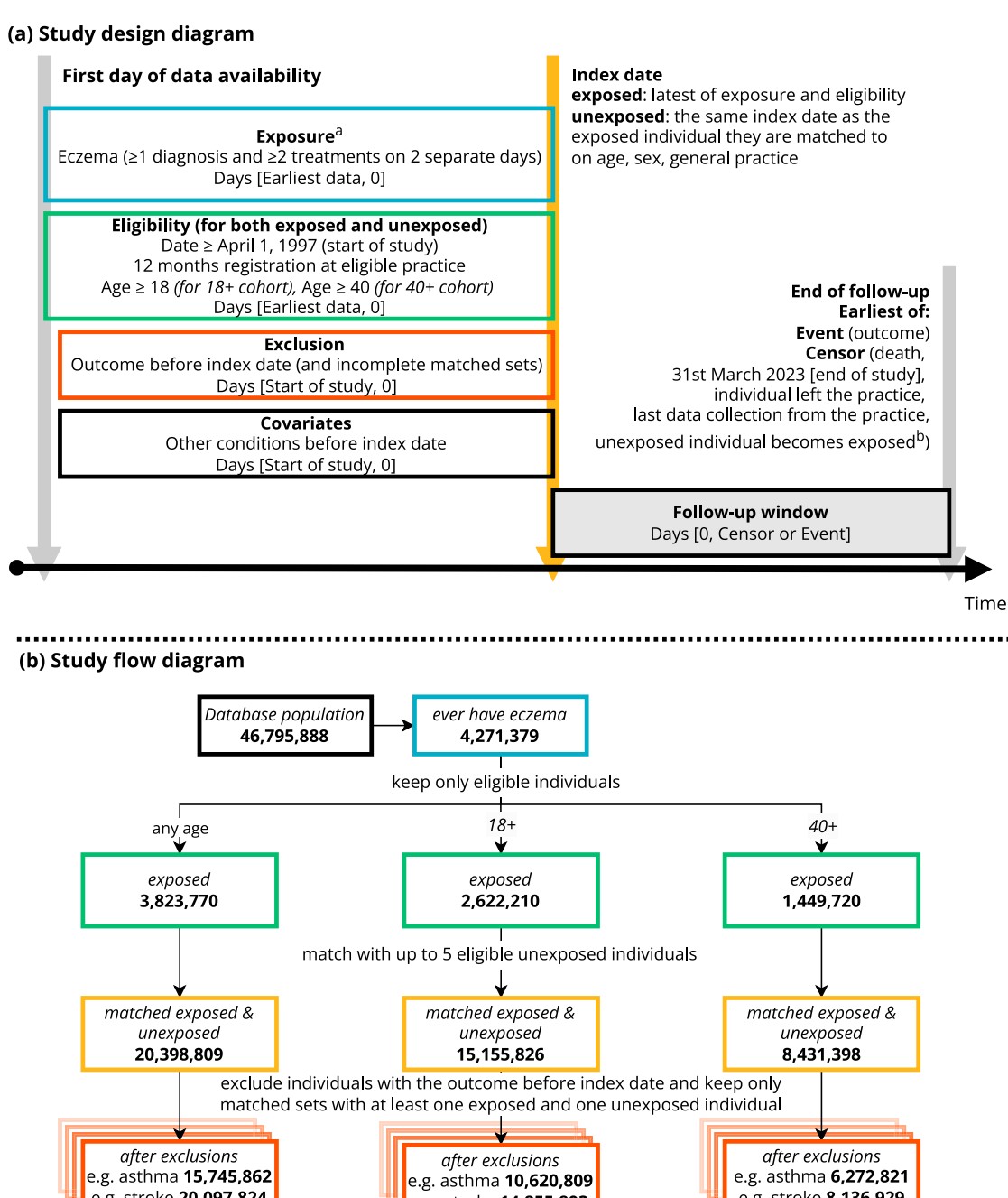

**(a) Study design diagram**

**First day of data availability**

**Index date**
**exposed**: latest of exposure and eligibility
**unexposed**: the same index date as the exposed individual they are matched to on age, sex, general practice

**Exposure**[a]
Eczema (≥1 diagnosis and ≥2 treatments on 2 separate days)
Days [Earliest data, 0]

**Eligibility (for both exposed and unexposed)**
Date ≥ April 1, 1997 (start of study)
12 months registration at eligible practice
Age ≥ 18 *(for 18+ cohort)*, Age ≥ 40 *(for 40+ cohort)*
Days [Earliest data, 0]

**End of follow-up**
Earliest of:
**Event** (outcome)
**Censor** (death,
31st March 2023 [end of study],
individual left the practice,
last data collection from the practice,
unexposed individual becomes exposed[b])

**Exclusion**
Outcome before index date (and incomplete matched sets)
Days [Start of study, 0]

**Covariates**
Other conditions before index date
Days [Start of study, 0]

**Follow-up window**
Days [0, Censor or Event]

Time

**(b) Study flow diagram**

*Database population*
**46,795,888**
→
*ever have eczema*
**4,271,379**

keep only eligible individuals

any age | 18+ | 40+

*exposed* **3,823,770** | *exposed* **2,622,210** | *exposed* **1,449,720**

match with up to 5 eligible unexposed individuals

*matched exposed & unexposed* **20,398,809** | *matched exposed & unexposed* **15,155,826** | *matched exposed & unexposed* **8,431,398**

exclude individuals with the outcome before index date and keep only matched sets with at least one exposed and one unexposed individual

*after exclusions* e.g. asthma **15,745,862** e.g. stroke **20,097,824** | *after exclusions* e.g. asthma **10,620,809** e.g. stroke **14,855,893** | *after exclusions* e.g. asthma **6,272,821** e.g. stroke **8,136,929**

**Fig. 1 | Study design and flow diagram. a** Study design diagram and (**b**) Study flow diagram, colour-coded by step. [a]Treatments include emollients, topical glucocorticoids, topical calcineurin inhibitors, systemic immunosuppressants (azathioprine, methotrexate, ciclosporin, mycophenolate), and oral glucocorticoids. [b]Unexposed individuals are censored on the day they meet the eczema diagnostic algorithm themselves, and can then be re-matched, this time as exposed individuals.

our results from analyses by eczema severity were generally similar to results from these studies (Supplementary Fig. 2).

## Discussion

### Summary of the most relevant findings

Besides confirming associations with atopic conditions, immune-mediated skin conditions and skin infections, we found strong associations with Hodgkin's lymphoma, Crohn's disease, coeliac disease, ulcerative colitis, and autoimmune liver disease, albeit with relatively small absolute rate differences for these outcomes. More common, but less strongly associated, outcomes included irritable bowel syndrome, oesophagitis, gastro oesophageal reflux disease, thromboembolic disease, obesity, chronic obstructive pulmonary disease, gastritis and duodenitis and peripheral neuropathies. Our severity analyses also suggest that some outcomes may be primarily associated with severe eczema, and not all eczema, for example cardiovascular outcomes, osteoporosis and fractures.

### Discussion of findings by category

The largest associations were found with atopic and allergic conditions, urticaria and alopecia areata, which is already well known from clinical practice, and recognised in guidelines on awareness of eczema comorbidities[5]. We found evidence of a link with skin infection, which is also well known clinically, staphylococcus infection being a diagnostic criterion for eczema[9]. We also found an association with COPD, however the increased HR in the <18 cohort suggests

## Table 1 | Baseline characteristics

| Characteristic | Without eczema | With eczema |
|---|---|---|
| N (any-age cohort) | 16,756,329 (100%) | 3,642,480 (100%) |
| Female | 9,364,020 (56%) | 2,012,612 (55%) |
| Age at index date (median [IQR]) | 27 (7, 51) | 24 (5, 49) |
| Index date (median [IQR]) | 2011-12-05 (2006-02-10, 2017-03-17) | 2012-03-19 (2006-05-30, 2017-05-03) |
| Follow-up time (median [IQR]) | 4.5 (1.7, 9.7) | 5.5 (2.2, 11.1) |
| **Presence of condition before index date** | | |
| Asthma | 1,435,064 (8.6%) | 613,492 (17%) |
| Food allergy | 108,180 (0.6%) | 78,267 (2.1%) |
| Allergic rhinitis | 927,699 (5.5%) | 423,059 (12%) |
| Allergic conjunctivitis | 122,049 (0.7%) | 68,865 (1.9%) |
| Eosinophilic oesophagitis | 618 (<0.1%) | 268 (<0.1%) |
| Alopecia Areata | 27,564 (0.2%) | 12,463 (0.3%) |
| Urticaria | 354,359 (2.1%) | 164,798 (4.5%) |
| Anxiety | 1,267,675 (7.6%) | 366,602 (10%) |
| Depression | 1,842,907 (11%) | 512,200 (14%) |
| Alcohol abuse | 124,518 (0.7%) | 34,589 (0.9%) |
| Cigarette smoking | 4,245,208 (25%) | 1,019,708 (28%) |
| ADHD | 54,464 (0.3%) | 14,514 (0.4%) |
| Autism | 52,959 (0.3%) | 18,167 (0.5%) |
| Hypertension | 1,544,517 (9.2%) | 374,224 (10%) |
| Coronary artery disease | 763,739 (4.6%) | 198,518 (5.5%) |
| Peripheral artery disease | 91,614 (0.5%) | 27,181 (0.7%) |
| Myocardial infarction | 150,174 (0.9%) | 37,460 (1.0%) |
| Stroke | 126,536 (0.8%) | 32,701 (0.9%) |
| Heart failure | 129,099 (0.8%) | 34,976 (1.0%) |
| Thromboembolic diseases | 128,014 (0.8%) | 39,049 (1.1%) |
| Obesity | 460,597 (2.7%) | 137,116 (3.8%) |
| Dyslipidaemia | 614,101 (3.7%) | 158,514 (4.4%) |
| Diabetes mellitus | 555,912 (3.3%) | 139,191 (3.8%) |
| Metabolic syndrome | 3201 (<0.1%) | 943 (<0.1%) |
| Hip fracture | 52,153 (0.3%) | 13,118 (0.4%) |
| Pelvis fracture | 22,116 (0.1%) | 5753 (0.2%) |
| Spine fracture | 33,260 (0.2%) | 8940 (0.2%) |
| Wrist fracture | 348,118 (2.1%) | 90,203 (2.5%) |
| Osteoporosis | 187,393 (1.1%) | 49,981 (1.4%) |
| Molluscum contagiosum | 174,281 (1.0%) | 105,982 (2.9%) |
| Impetigo | 470,683 (2.8%) | 239,225 (6.6%) |
| Herpes simplex | 225,737 (1.3%) | 92,266 (2.5%) |
| Dermatophyte infection | 741,793 (4.4%) | 339,711 (9.3%) |
| Cutaneous warts | 943,423 (5.6%) | 352,254 (9.7%) |
| Lung cancer | 10,881 (<0.1%) | 2910 (<0.1%) |
| Breast cancer | 88,755 (0.5%) | 21,587 (0.6%) |
| Prostate cancer | 61,022 (0.4%) | 14,654 (0.4%) |
| Pancreatic cancer | 1592 (<0.1%) | 426 (<0.1%) |
| Non-Hodgkin lymphoma | 14,631 (<0.1%) | 3955 (0.1%) |
| Hodgkin lymphoma | 3915 (<0.1%) | 1138 (<0.1%) |
| Myeloma | 4006 (<0.1%) | 879 (<0.1%) |
| CNS cancers | 12,842 (<0.1%) | 3367 (<0.1%) |
| Melanoma | 49,225 (0.3%) | 13,070 (0.4%) |
| Nonmelanoma skin cancer | 185,337 (1.1%) | 48,804 (1.3%) |
| Alzheimer's dementia | 48,512 (0.3%) | 11,132 (0.3%) |
| Vascular dementia | 25,060 (0.1%) | 5896 (0.2%) |
| Abdominal hernia | 417,179 (2.5%) | 112,260 (3.1%) |

## Table 1 (continued) | Baseline characteristics

| Characteristic | Without eczema | With eczema |
|---|---|---|
| Appendicitis | 157,025 (0.9%) | 39,270 (1.1%) |
| Autoimmune liver disease | 4956 (<0.1%) | 1415 (<0.1%) |
| Barrett's oesophagus | 29,775 (0.2%) | 8097 (0.2%) |
| Cholecystitis | 45,793 (0.3%) | 12,195 (0.3%) |
| Coeliac disease | 29,173 (0.2%) | 9653 (0.3%) |
| Crohn's disease | 24,664 (0.1%) | 9436 (0.3%) |
| Diverticular disease | 224,230 (1.3%) | 62,132 (1.7%) |
| Fatty liver | 60,169 (0.4%) | 18,902 (0.5%) |
| Gastritis and duodenitis | 295,187 (1.8%) | 92,550 (2.5%) |
| Gastro oesophageal reflux | 667,375 (4.0%) | 211,087 (5.8%) |
| Irritable bowel syndrome | 407,739 (2.4%) | 132,643 (3.6%) |
| Fibrosis/sclerosis/cirrhosis | 16,723 (<0.1%) | 5142 (0.1%) |
| Oesophageal varices | 3951 (<0.1%) | 1161 (<0.1%) |
| Oesophagitis | 305,002 (1.8%) | 95,584 (2.6%) |
| Pancreatitis | 29,603 (0.2%) | 7853 (0.2%) |
| Peptic ulcer disease | 82,341 (0.5%) | 22,072 (0.6%) |
| Peritonitis | 11,154 (<0.1%) | 2804 (<0.1%) |
| Ulcerative colitis | 36,104 (0.2%) | 12,201 (0.3%) |
| Epilepsy | 129,082 (0.8%) | 37,357 (1.0%) |
| Migraine | 530,815 (3.2%) | 159,329 (4.4%) |
| Multiple sclerosis | 19,054 (0.1%) | 5021 (0.1%) |
| Parkinson's disease | 29,793 (0.2%) | 6372 (0.2%) |
| Peripheral neuropathies | 784,610 (4.7%) | 227,252 (6.2%) |
| COPD | 228,151 (1.4%) | 68,681 (1.9%) |
| Oral glucocorticoids | 1,072,867 (6.4%) | 481,055 (13%) |
| Systemic immunosuppressants | 83,960 (0.5%) | 26,634 (0.7%) |
| **N (18+ cohort)** | 12,588,779 (100%) | 2,567,047 (100%) |
| Age at index date (median [IQR]) | 37 (23, 58) | 37 (23, 59) |
| Follow-up time (median [IQR]) | 4.2 (1.7, 8.9) | 4.8 (1.9, 9.9) |
| **N (40+ cohort)** | 7,002,612 (100%) | 1,428,786 (100%) |
| Age at index date (median [IQR]) | 56 (43, 69) | 56 (43, 70) |
| Follow-up time (median [IQR]) | 5.6 (2.4, 10.5) | 6.2 (2.7, 11.3) |

ADHD Attention deficit hyperactivity disorder, CNS Central nervous system, COPD Chronic obstructive pulmonary disease.
Numbers are N (%) unless otherwise indicated.

that there may be overlap with asthma recordings, as COPD usually occurs in older age.

We found evidence for associations with autoimmune liver disease and liver fibrosis/sclerosis/cirrhosis, albeit with small rate differences, and fatty liver, with a larger, but still relatively small, rate difference. Sensitivity analyses suggest that at least some of the effect seen may be explained by consultation bias, especially for fatty liver. We saw dose-response relationships with eczema severity and very large HRs for severe eczema, suggesting that most of the increased risk may be in those with severe eczema. We found little existing evidence on these associations, so it is likely there was little awareness of these potential links, however given the relatively small rate differences these outcomes may be less important to consider in screening and prevention contexts.

We found strong evidence for associations with inflammatory bowel diseases, that held up in sensitivity analyses. We also saw risk increasing with more severe eczema, with some of the largest effects for severe eczema seen across all outcomes. A recent study from UK population data showed similar results[10], and other studies have

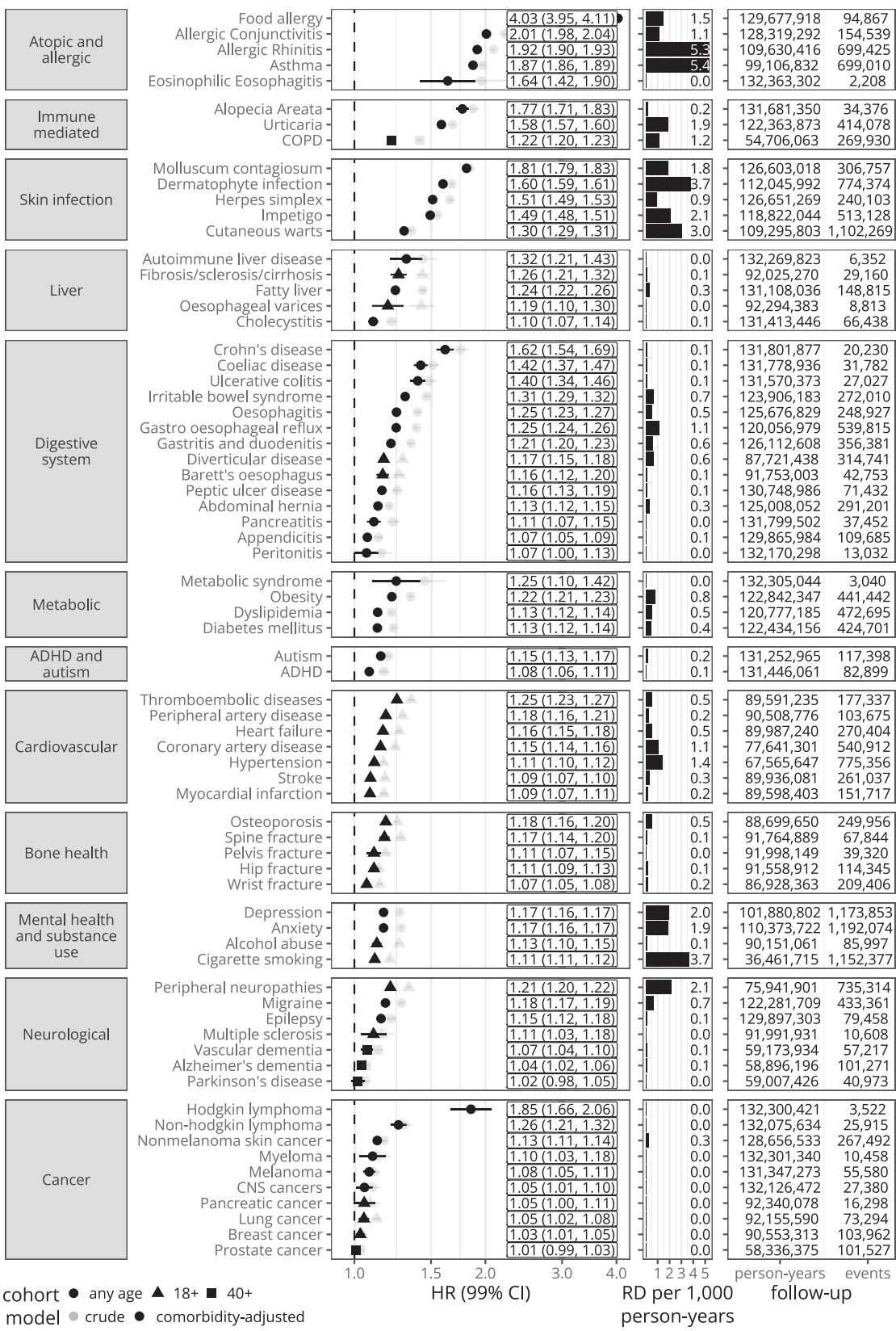

**Fig. 2 | Main results - Eczema compared to no eczema.** Hazard ratios (HR) with 99% confidence intervals (99%CI) (represented by vertical bars) from Cox regression, estimated absolute rate difference per 1000 person-years (RD per 1000 p-years) (rate in those with eczema – estimated rate in those without eczema; the rate in those without eczema is estimated as the rate in the exposed * [1/hazard ratio]), person-years and number of events. Hazard ratios in labels are from adjusted models. Estimates and counts are from the respective main cohort after excluding individuals with the respective outcome before index date (which explains differences in follow-up time between outcomes). ADHD Attention deficit hyperactivity disorder, CNS Central nervous system, COPD, Chronic obstructive pulmonary disease.

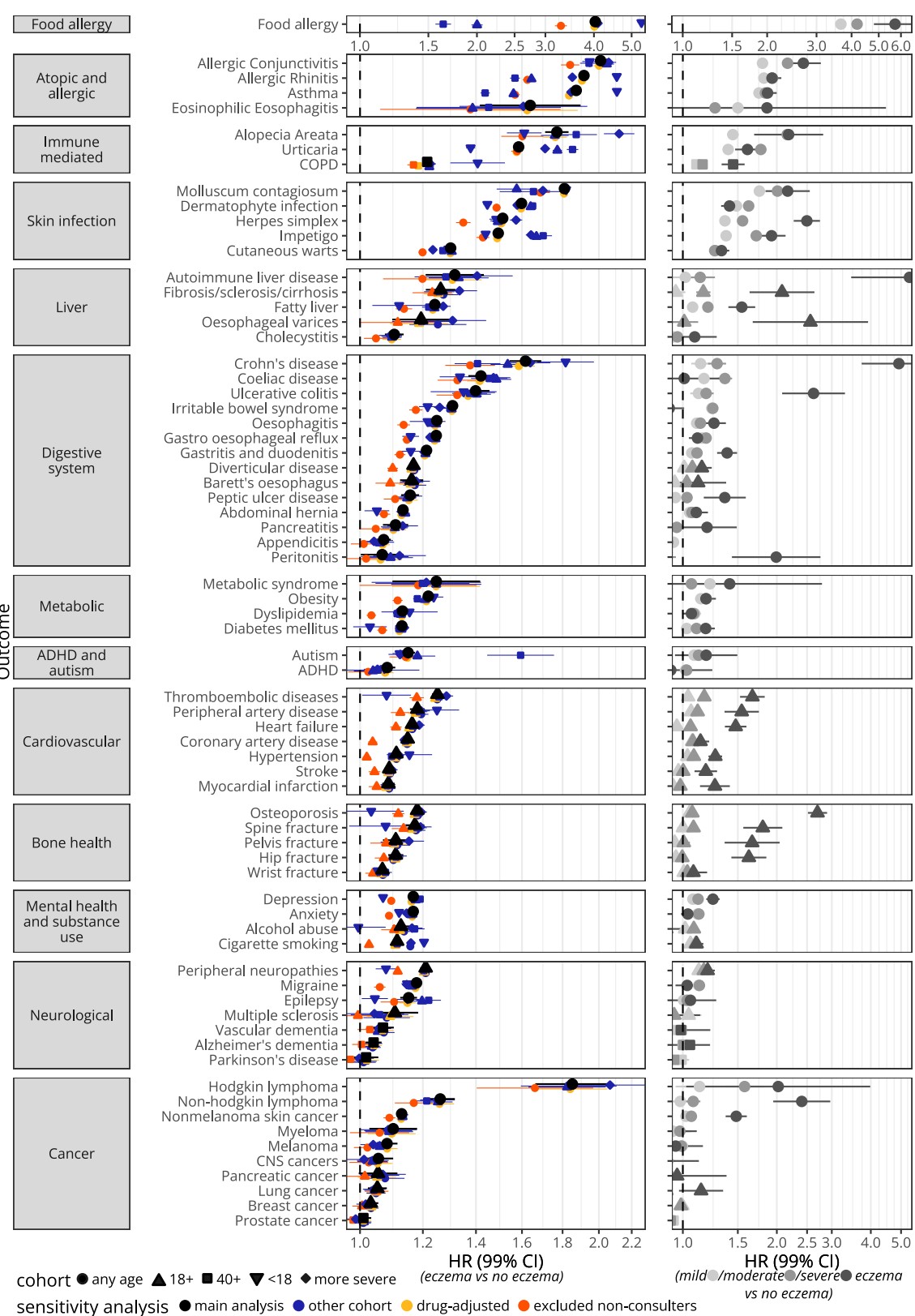

similarly concluded that a, possibly bidirectional, association exists[11]. The comparison with other outcomes in our study suggests that inflammatory bowel diseases may be some of the most relevant to consider for future research, however, small rate differences may suggest less relevance in informing screening or prevention measures in people with eczema.

We also found evidence for diseases of the oesophagus, albeit with less clear dose-response relationships with eczema severity. Some previous population-based studies have shown similar results, e.g., for gastro-oesophageal reflux[12], however, findings may be partially explained by an increased risk of developing eosinophilic oesophagitis, for which awareness is increasing but may still be misdiagnosed[13].

**Fig. 3 | Results from sensitivity and secondary (severity) analyses.** Hazard ratios (HR) with 99% confidence intervals (99%CI) (represented by vertical bars) from Cox regression. Left: Comorbidity-adjusted results from the respective main cohort (in black), comorbidity-adjusted results from analyses where the other cohorts were used (in blue), results from additionally drug adjusted models (in yellow), and comorbidity-adjusted results from analyses excluding non-consulters from the respective main cohort (in orange). In the more severe cohort, individuals are considered exposed when they had an additional record indicating more severe eczema (phototherapy, or prescriptions for potent topical corticosteroids, topical calcineurin inhibitors or systemic immunosuppressants) after having met the eczema diagnosis algorithm (i.e., the comparators matched to these exposed individuals also included individuals with eczema considered to be less severe). Person-years and number of events are given in Supplementary Tables 2 and 3 for each outcome. Right: Additionally hospital-admission adjusted comorbidity-adjusted hazard ratios from Cox regression comparing those with mild eczema, moderate eczema, and severe eczema to those without eczema. Person-years and number of events are given in Supplementary Table 4 for each outcome. Both left and right: Results for food allergy are displayed with their own x-axis since HRs were considerably higher than for any other outcome.

Of other digestive system conditions studied, the evidence of association was strongest for coeliac disease, which has previously been studied together with other autoimmune conditions[14]. For Irritable bowel syndrome, and gastritis and duodenitis, the evidence of association was also relatively strong considering strength of associations, and results from sensitivity and severity analyses.

Our results suggest small relative, but potentially considerable absolute, increased risks for depression and anxiety; for alcohol abuse and cigarette smoking the evidence was weaker.

Our findings suggest uncertain evidence and/or weak associations with autism and attention deficit hyperactivity disorder (ADHD) in line with existing guidelines[5]. Our findings for autism in particular should be interpreted with caution as results from analyses where the 40+ cohort were used showed a large increase in the hazard ratio, which is unexpected, given autism is a usually diagnosed in childhood, and it is unlikely people with eczema would have higher rates of autism in adulthood.

We found a somewhat increased risk of thromboembolic (e.g., deep vein thrombosis, phlebitis) and peripheral artery disease, with weaker evidence for heart failure, coronary artery disease and hypertension, and only very weak, or for a very small increased risk, for stroke and myocardial infarction. For some conditions, e.g., hypertension, increased risk may be almost entirely explained by consultation bias. We saw relatively large HRs for severe eczema for cardiovascular outcomes, as was the case in a previous population-based study[3].

While we saw an association with obesity, we did not see a dose-response relationship with eczema severity, and again saw that consultation bias may be an important explanatory factor; this was similar for metabolic syndrome albeit with few events occurring in our study. While for diabetes we saw risk increasing with more severe eczema, in the main analysis the risk was relatively small, possibly explained by consultation bias, and we saw a null result when using the <18 cohort. This may suggest that eczema has little or no effect on diabetes occurring in younger age, but may still have an effect on diabetes occurring in older age.

In existing guidelines, the association with osteoporosis was graded as being of high certainty[5], based on three studies[15,16], one population-based matched cohort study from Taiwan[17] showing HRs of more than 4 (as compared to our HR of 1.18 [1.16-1.20]). While in our study there was evidence of only small increases of risk for osteoporosis and fractures (compared to other outcomes), we found relatively large HRs for severe eczema, as was the case in a previous study[7], suggesting risk may potentially only be increased in those with severe eczema.

We found a relatively large relative and absolute effects for peripheral neuropathies, about half of the records that made up this outcome being for sciatica. There was also some evidence for an association with migraine, a recent study showing similar effect sizes (HR from fully adjusted model 1.2 [1.2–1.26]) to ours (aHR 1.18 [1.17–1.19])[18]. However, sensitivity analyses suggest these associations may be explained in large part by consultation bias.

Our findings are consistent with those from a previous study that showed no evidence for association with solid organ cancers but associations with lymphomas[8]. The larger sample size of our study allowed more precise estimation of the association with Hodgkin's lymphoma, which has one of the largest effect estimates of any outcomes, but a low absolute difference.

## Strengths

Our study has several strengths, including the use of the large and representative CPRD Aurum database, meaning our results are likely to be generalizable to the general population of England[3,4,7,8,19,20].

The approach we took to study multiple outcomes has advantages compared to traditional approaches, the most obvious benefit being vastly increased efficiency and speed of evidence generation. The results for each outcome are also directly comparable to each other, providing the opportunity to put results in context with outcomes that are well known to be linked to eczema (e.g., food allergy), and outcomes that are unlikely to be linked to eczema (e.g., cancer), acting as positive and negative controls respectively. This may be particularly useful when interpreting and comparing absolute rate differences across outcomes, which may help in judging public health impact of interventions.

We used a strategy for confounding-adjustment, the suitability of which to produce correct confounder-adjusted effect estimates across multiple outcomes has been previously described theoretically and demonstrated practically[21]. Requirements, including large sample sizes and information on a large number of variables (and their timing) that may confound the association between the exposure and any of the outcomes, is met by our data source and large study population. Our results were almost identical to those from four previous CPRD GOLD studies, for which dedicated strategies to adjust for confounding were developed, suggesting our approach is broadly suitable for producing confounder-adjusted estimates across multiple outcomes. Our results were more conservative than from studies done across a range of other data sources and designs[5], suggesting effects may have often been over-estimated in the past, possibly due to inadequate adjustment for confounding[5].

Consistently running analyses for all cohorts across all outcomes, provides the opportunity for closer inspection when results are considerably different. For example, while for many outcomes the use of other cohorts did not change results, for atopic and allergic conditions effect estimates decreased with cohort age cut-off; a finding that makes sense clinically, as eczema in childhood may be more strongly linked to allergies than in adulthood. An additional advantage of consistently having results available from the any-age cohort for all outcomes, is that the any-age and <18 cohorts included only newly-recorded eczema, while the 18+ and 40+ cohorts included both newly-recorded and previously-recorded eczema, and therefore acted as a further sensitivity analysis.

The sensitivity analysis excluding non-consulters revealed that consultation bias may be an issue across all outcomes in studies on eczema using primary care data. Nevertheless, results suggest that many associations cannot fully be explained by consultation bias. Analyses by eczema severity can further strengthen the evidence of a given outcome by providing evidence of a dose-response relationship, or highlight when risk may primarily be increased in those with severe eczema.

The cross-outcome approach has additional strengths that may help avoid researcher biases. Firstly, in epidemiological research, hundreds of tests across multiple studies are often performed using the same data source. However, multiple testing is rarely considered since these tests are done across many different studies. In our study it was straightforward to include adjustments for multiple testing (although this was less important to consider given the large sample size supplied high power to test multiple outcomes). We indicate whether results are significant under Bonferroni-correction in Supplementary Tables 1–4. Secondly, our approach limits the possibility that study design choices and covariate selection were made to explicitly increase or decrease the results for a particular outcome by necessitating that one study design, including all variations on cohort composition and covariate sets, was applied to, and reported for all outcomes.

### Limitations

Our study has limitations. We were not able to account for missing data, given that there are no explicitly missing values. We defined outcomes using primary care only, which may miss diagnoses only captured in other care settings, or for diagnosis for which an individual does not consult. Ascertainment in primary care is better for some conditions than for others. While this is not a concern for eczema, as almost all eczema is managed in primary care[22], some outcomes (e.g., especially those that are acute and serious such as myocardial infarction) are mostly managed in secondary care, and despite feedback from secondary to primary care, some of these diagnoses might be missed. While linking to secondary care data may have helped address some of these issues, this would have come at the cost of reducing sample size, length of follow-up and generalisability of CPRD data[23].

Despite adjusting each analysis for a large range of potential confounders, as in all observational studies, residual and unmeasured confounding cannot be excluded. We did not explicitly adjust for deprivation (e.g., using the index of multiple deprivation which can be linked to CPRD Aurum data) however given it is based on small area units with an average of 1,600 residents from the 2011 census[24], it may not provide better adjustment for deprivation than is already achieved by matching on GP practice. We did not adjust for ethnicity, as the proportion of missing ethnicity data may have introduced selection bias. Most previous studies that did adjust for ethnicity found little difference to main results when additionally adjusting for ethnicity[3,4,8,20]. Future research may consider more detailed investigations of the role of ethnicity, not just as a confounding factor. There may be residual confounding through lifestyle factors, not captured in our diagnosis-based smoking, obesity and alcohol abuse definitions. By excluding individuals with the outcome of interest before index date we aimed to minimise reverse causation, however, reverse causation may still partially explain findings as timing of diagnoses in EHRs may not accurately represent the actual start of conditions.

There may also be limitations relating to defining eczema. Our eczema definition was based on a validated algorithm, and we ran sensitivity analyses with a cohort of individuals with more severe, and therefore likely more definite eczema. However, eczema may still be difficult to establish in primary care records (as individuals, particularly those with milder disease, may not consult for their symptoms), or even in clinical practice itself, and our exposed group may include different subtypes of eczema (which may be associated with different sets of outcomes).

We were limited to defining eczema severity based on prescriptions and hospital admissions, which can only approximate severity. We therefore encourage cautious interpretation of effect estimates, and that results should be used as an additional tool to judge the strength of evidence, rather than representing precise estimates of the increase in risk in people with mild, moderate, or severe eczema.

Our approach of exploring multiple outcomes may also have limitations compared to studies focused on a narrower set of outcomes. Such studies may reveal more about the mechanisms behind associations, for example, by considering which individual variables confound, mediate, or modify the association and may benefit from more detailed application of expert knowledge, including reviews of the existing literature and discussion of biological plausibility, to each exposure-outcome relationship.

### Interpretation

Our results can be used to judge the plausibility and strength of links between eczema and the subsequent development of adverse health outcomes. Rather than relying solely on a statistical significance threshold, which due to high power may be met for unimportant or small effects, the strength of association together with findings from sensitivity and severity analyses should be considered. Absolute measures of effect allow judging the potential public health relevance. Uniquely, across all analyses our study offers a comparison between outcomes. Whether associations are causal, implying effective diagnosis and treatment for eczema could prevent the development of these comorbidities, is not possible to determine from this study alone. However, irrespective of causality, the increased risk found for being diagnosed with conditions subsequent to eczema emphasises the importance of a multidisciplinary approach to care for these individuals. Future research may aim to investigate mechanisms through which eczema, and especially more severe eczema, may be associated with outcomes, such as sleep deprivation, medications, low self-esteem[1], common causes of eczema and outcomes such as atopy, the role of eczema as a systemic disorder associated with systemic inflammation, and the extent to which outcomes are shared with other chronic inflammatory conditions (e.g., psoriasis)[25]. Replicating this work in other large population-based data sources will further strengthen the evidence.

In conclusion, we give a comprehensive overview of adverse health outcomes associated with eczema, for each of the 71 outcomes providing evidence from a large and representative database, including from sensitivity analyses and analyses by eczema severity. The cross-outcome approach offers additional benefits of comparability between results, reduced investigator biases and efficiency of evidence generation. Findings can be used to inform guidelines and clinical practice, and as a baseline for more detailed research on individual outcomes.

## Methods

### Ethics

The study was approved by the London School of Hygiene & Tropical Medicine Research Ethics Committee (Reference number: 29781). This study is based on data from the CPRD obtained under license from the U.K. Medicines and Healthcare products Regulatory Agency. The data are provided by patients and collected by the National Health Service (NHS) as part of their care and support. The study was approved by the Independent Scientific Advisory Committee (Protocol reference number: 23_002665). Individual patient consent is not required or possible since CPRD provides anonymised data.

### Study design and setting

We used a matched cohort study design with deidentified routinely collected UK primary care electronic health records (EHR) data (April 1st 1997, to March 31st 2023) from CPRD Aurum, which includes over 46 million people, and has been found to be representative of the general population of England in terms of age, sex, geographical spread and deprivation[19].

## Study population

We created different cohorts based on minimum age at inclusion (Fig. 1). For all cohorts, we used an algorithm to identify individuals with eczema (at least one record of an eczema diagnostic code and at least two records for eczema therapies [emollients, topical glucocorticoids, topical calcineurin inhibitors, oral glucocorticoids or systemic immunosuppressants] on two separate days). An analogous algorithm has been previously validated in UK primary care data and was found to have a positive predictive value of 86%[26]. We then included individuals in the eczema exposed group on the latest of: (1) Date they met the eczema definition; (2) One year since practice registration (to allow us to reliably capture baseline health status); (3) Study start (April 1, 1997); and (4) 18th (18+ cohort) or 40th birthday (40+ cohort), or no age limitation (any-age cohort). For the 18+ and 40+ cohorts, meeting the eczema definition could occur before individuals became eligible (i.e., individuals with both new and existing eczema were included, a recommended approach for relapsing conditions like eczema to better assess longer-term effects of an exposure)[27].

Eczema exposed individuals were matched (without replacement) to up to 5 unexposed individuals with at least 1-year prior registration, on age (2-year calliper), sex, and general practice in calendar date order. The index date for comparators was set to the index date of the exposed individual they were matched to. Comparators were censored on the day they met the eczema definition themselves, and could then be re-matched, this time as exposed individuals. Individuals were followed up until the date of outcome, or until they were censored (death, left practice, or for comparators, when they met the eczema definition). For each outcome-specific analysis, individuals who had the outcome before their index date were excluded (Fig. 1).

For sensitivity analyses, we created two additional cohorts. Firstly, a (more severe) cohort where individuals were only considered exposed when they had an additional record indicating more severe eczema after having met the eczema definition (i.e., the comparators matched to these exposed individuals also included individuals with eczema considered to be less severe). Records indicating more severe eczema included records for phototherapy, or prescriptions for potent topical corticosteroids, topical calcineurin inhibitors or systemic immunosuppressants (azathioprine, methotrexate, ciclosporin, mycophenolate)[3,4,7,8,20]. Secondly, a subset of the any-age cohort of individuals that met the eczema definition before their 18th birthday (<18 cohort).

In an additional sensitivity analysis performed to address consultation bias, using each outcome's respective main cohort, individuals who did not have any of four common records (9N11.00 - Seen in GP's surgery, 22 A..00 - Body weight, 4....00 - Laboratory test, 246..00 - O/E - blood pressure reading) in the year prior to index date were excluded ("analysis excluding non-consulters").

For secondary analyses, eczema severity was defined as mild, moderate, or severe as a time-updated variable. People with eczema were assumed to have mild disease in the absence of any evidence for moderate or severe disease. When they received prescriptions for potent topical corticosteroids or topical calcineurin inhibitors (after meeting the eczema definition), they were considered as having moderate eczema. When they had eczema recorded in secondary care, received prescriptions for systemic immunosuppressants (azathioprine, cyclosporine, methotrexate, or mycophenolate mofetil), or had phototherapy recorded, they were considered as having severe eczema. Individuals' assigned severity could progress from mild to moderate to severe, but not revert from moderate or severe to mild, or from severe to moderate; hence this variable denoted whether a person had ever-experienced moderate or severe eczema, which is an approach used in several previous studies[3,4,7,8,20]. To limit the the effect of reverse causation (i.e., individuals may have eczema recorded in secondary care when they are admitted to hospital due to another condition), we additionally adjusted for time-updated hospital admission. We present results from these additionally hospital-admission-adjusted models in the Results section, and in Supplementary Table 4 together with comorbidity-adjusted results. Severity analyses are limited to those eligible for linkage to secondary care records.

## Outcomes

We included all adverse health outcomes (except those defined by death) on which statements were released in the AAD guidelines on comorbidities for adults with eczema[5], covering a wide range of atopic and allergic, immune-mediated, mental health and substance use, cardiovascular, metabolic, bone health and skin infection outcomes. We also included outcomes that had been previously studied in relation to eczema (i.e., cancers, dementia)[8,28] or which had been identified as an area of particular interest by previous hypothesis-generating work (i.e., digestive system, neurological conditions)[6]. We used code lists and algorithms from previous studies[3,4,7,8,20,29,30] and mapped these to CPRD Aurum medical and product codes (code lists available in the study repository). The most commonly occurring codes for each outcome are in Supplementary Table 1.

## Statistical analysis

We presented descriptive statistics of the cohorts at baseline by eczema status. We used Cox proportional hazards regression, stratifying on matched set, to estimate hazard ratios (HRs) for the effect of eczema on each outcome. For each analysis, we estimated minimally-adjusted (implicitly adjusted through matching on age, sex and general practice, and calendar time, as comparators entered the cohort on the same day as exposed individuals) and comorbidity-adjusted HRs (additionally adjusted for history of each other outcome at baseline). As sensitivity analysis, we also estimated drug-adjusted (additionally adjusting the comorbidity-adjusted model for oral corticosteroids and systemic immunosuppressants, defined as history of at least one prescription at the index date) HRs, to account for drugs that are sometimes used in eczema treatment but are more commonly used in the treatment of other conditions. We also calculated crude rate differences and estimated adjusted rate differences based on the hazard ratio (as the rate in those without eczema times the inverse of the hazard ratio subtracted from the rate in those with eczema).

The validity of our confounding adjustment strategy for multiple outcomes has been previously described. In summary, covariates that are causes of either the exposure or of any outcome are adjusted for, which in our study includes baseline values of all outcomes and other pre-exposure covariates (i.e., age, sex, general practice)[21].

To account for multiple testing, we reported wider 99% instead of usual 95% confidence intervals. While in our interpretations we do not rely on significance cut-offs, we have additionally reported whether effect estimates were significant for each outcome under Bonferroni correction when counting all outcomes (with 71 outcomes considered, estimates would be considered significant under Bonferroni correction with a p-value less than $0.05/71 = 0.0007$) in Supplementary Table 1.

To benchmark results from our study against results from studies specifically designed to assess the risk of certain outcomes, we report whether our results were similar to those from four previous CPRD GOLD studies[3,4,7,8].

## Pipeline

For all 71 outcomes, we ran analyses for all cohorts (any-age, 18+, 40+, more severe, <18), for all three models (minimally-adjusted, comorbidity-adjusted, drug-adjusted) and for the respective main cohort we ran analyses excluding non-consulters. We considered our primary results to be those from comorbidity-adjusted models, and from the age cohort that was most relevant to the typical age of onset for each

given outcome (e.g. the any-age cohort for asthma, the 18+ cohort for hypertension, the 40+ cohort for dementias - a full specification for each outcome is listed in Supplementary Table 1). We considered the following as sensitivity analyses for given outcomes: (1) results from both minimally- and drug-adjusted models; (2) from the cohorts with the other minimum/maximum ages at inclusion; (3) the more-severe cohort; (4) the analyses excluding non-consulters.

We used R version 4.3.1 and organised the research pipeline using the targets R package. Each analysis and data management step was represented by a single function that was mapped across all combinations of outcomes, cohorts and models, ensuring reproducibility of the computationally expensive pipeline[31].

### Reporting summary

Further information on research design is available in the Nature Portfolio Reporting Summary linked to this article.

## Data availability

Data supporting the findings of this study are available in the article and its Supplementary information. The data underlying this article is provided by the UK CPRD electronic health record database, which is only accessible to researchers with protocols approved by the CPRD's independent scientific advisory committee. Data access may incur a cost and further details can be found here: https://www.cprd.com/data-access.

## Code availability

All analysis code and codelists used for this study are available at https://doi.org/10.5281/zenodo.10649715.

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

## Acknowledgements

This work uses data provided by patients and collected by the NHS as part of their care and support. This work was funded by a Wellcome Trust Senior Research Fellowship in Clinical Science (205039/Z/16/Z) awarded to Sinéad M. Langan. Krishnan Bhaskaran is funded by a Wellcome Senior Research Fellowship (220283/Z/20/Z). Helen Strongman is funded by the National Institute for Health Research (NIHR) though an Advanced Fellowship (NIHR301730). The views expressed in this publication are those of the author(s) and not necessarily those of the NIHR, NHS or the UK Department of Health and Social Care.

## Author contributions

JM contributed to Conceptualization, Data curation, Formal Analysis, Funding acquisition, Investigation, Methodology, Project administration, Resources, Software, Supervision, Validation, Visualization, Writing – original draft, and Writing – review & editing. SML contributed to Methodology, Supervision, Validation, and Writing – review & editing. AS contributed to Methodology, Validation, and Writing – review & editing. JM, AS, HS, KB, AR, SD, KEM, and SML contributed to Methodology and Writing – review & editing.

## Competing interests

Julian Matthewman, Anna Schultze, Kathryn E Mansfield, Spiros Denaxas, Krishnan Bhaskaran, Amanda Roberts, and Helen Strongman have no conflicts of interest to report relating to the findings. Sinéad M Langan is a co-investigator in a consortium with industry and multiple academic partners (BIOMAP-IMI.eu) but is not in receipt of industry funding.
