## [Transparent Peer Review file · Nature Communications]

Cohort studies on 71 outcomes among people with atopic eczema in UK primary care data

Corresponding Author: Dr Julian Matthewman

Version 0:

Reviewer comments:

Reviewer #1

(Remarks to the Author)

Reviewers: Hywel C. Williams and Lloyd Steele

General comments

- This submission represents a large body of work produced by international experts in the field of routine healthcare records.
- Importantly, the team also have content expertise in atopic eczema.
- We liked the general epidemiological approach of investigating multiple outcomes in the same study as an efficient and cost-effective way of addressing multiple questions.
- Much thought has been given to sensitivity analyses.
- The work is also well reported.
- The study will be a useful benchmark for other national databases to try and replicate

Scientific credibility

- There is a hint in the writing style of the manuscript that the many studies conducted to date have been a waste of time “The existing evidence base, encompassing hundreds of studies employing heterogeneous approaches to study design, analysis and data management, hinders comparison, and has been slow and expensive to build”. Sometimes, consistency from a range of study designs is a strength. Sometimes doing things slowly taking into account previous research can also be a strength.
- Abstract: The chronology of the inception disease (eczema) and subsequent outcomes is unclear in the abstract. Asthma and allergic rhinitis (the most obvious associated diseases that tend to develop later in the so called allergic march) are not mentioned in the abstract.
- Severity. Is there any evidence for the approach for severity? A single prescription of 30g Betnovate would presumably lead to a diagnosis of “moderate-severe” eczema, but with many adults included this could be for hand eczema? In the validation approach above, we see that none of the 200 patients had severe disease and only 25% had symptoms of eczema in the past year (which seems like a very mild cohort).
- Distribution of age at index date and follow-up. The median age of 24 for index case in eczema seemed high. Can you show the distribution of this? We also found it difficult to link the long-term outcomes (e.g. dementia) with the relative short follow-up period, so it would also be useful to see the distribution of the follow-up time.
- Most of the associations are not novel eg the links between atopic eczema and inflammatory bowel disease. We found 66 citations on this topic including a reasonably well conducted systematic review that confirmed the bidirectional association between atopic eczema and IBD (Lee H et al Bidirectional relationship between atopic dermatitis and inflammatory bowel disease: A systematic review and meta-analysis. J Am Acad Dermatol. 2020;83:1385-1394). The authors should state clearly what the really novel associations discovered in this important piece of work are in the abstract or perhaps state that the work is more confirmatory than novel in terms of the associations found.
- What is novel is the comprehensiveness and precision and representativeness of the associations and some degree of dose response in relation to severity based on associated treatments. The presentation of absolute risks is also valuable for estimating the magnitude of the health issues to individuals and public health planners.
- Given that eczema is such a common disease of childhood, it is odd that a children’s cohort has not been examined specifically (instead of being lumped into the all ages group), especially given increased confidence in the diagnosis of atopic eczema at this age (as opposed to irritant contact dermatitis, allergic contact dermatitis, varicose eczema, asteatotic eczema etc that are much more common as a differential diagnosis in adults that might be diagnosed in primary care as just “eczema” and treated with the same topical treatments as atopic eczema)
- Specificity: It is not clear how specific these findings are to atopic eczema – perhaps similar findings would emerge for other chronic inflammatory or infectious conditions (eg psoriasis, rheumatoid, periodontal disease) when examined in such a

way in a massive database. Comparing atopic eczema to the general population is a good start, but we would have liked to have seen some “active” controls too to minimise the possibility of generation of spurious associations.

- Perhaps the many outcomes in this study are shared by the family of conditions characterised by T helper cell 2 inflammatory responses – such a notion might be explored in some follow-on research.
- Fig 2 shows that all 71 conditions are significantly increased with atopic eczema (with the exception of prostate cancer and Parkinson’s disease which did not quite reach statistical significance). We appreciate this is probably because the team only studied those diseases that were rumoured in various guidelines and studies to be associated with eczema (so those rumours were right after all!), but we are left wondering whether the pattern in Fig 2 is an epiphenomenon of large routine health data research that examines just one disease? We would have found the specificity of the findings more convincing if the risk estimates had been more scattered and some were negative (eg previous studies have suggested a lower risk of brain cancer). We did not see any evidence of negative controls as hinted in the discussion as almost everything turned out to be positive.
- Significant results: It can be difficult to ascertain what is a positive result and negative result. Some examples are: Diabetes. The discussion reports “Our findings would support stating that eczema may not be associated with diabetes”. The results for diabetes show aHR 1.12 [99% interval 1.11-1.14] and RD 0.43 (which is a larger RD than reported for several diseases classed as significant). Myeloma. Eczema was weakly associated with myeloma (aHR 1.11 [1.04-1.19]; RD 0.01), We found evidence for associations with autoimmune liver disease and liver fibrosis/sclerosis/cirrhosis, albeit with small rate differences (results were aHR 1.35 [1.24-1.47]; RD 0.02) and (aHR 1.27 [1.22-1.33]; RD 0.09), respectively). Lymphoma is reported as a less common but more strongly associated outcomes, but the RD is only 0.02-0.05. It can be quite unclear where the line is drawn for diabetes not being associated but autoimmune liver disease being reported as a new significant association .
- Surveillance bias due to increased healthcare contact from the index condition is not mentioned at all in the submission.
- Multiple testing: The authors report “To account for multiple testing, we reported wider 99% instead of usual 95% confidence intervals”, but this is not quite the same as correcting for multiple testing. They report a Bonferonni corrected p-value, but every outcome in the minimally-adjusted model are below this p-value (that is, $p < 0.0007$) and in 65/71 of the comorbidity-adjusted model. While the authors rightly do not focus on significance here, multiple testing appears to be an issue.

Interpretation and clinical utility of the data

- Biological plausibility is often a retrospective exercise, and sometimes associations that do not appear plausible are found to have some plausibility at a later time when more is known about the disease. But some discussion on biological plausibility for the outcomes or groups of outcomes would be nice.
- We like the idea of using multiple cohorts within a national database of routine health data to sort of verify the consistency of findings, but it is not quite the same as replication in another routine health data set that has been set up in a different way. So we hope that this work will be replicated by others with the skills to interrogate other large routine health data sources such as Taiwan.
- Context to values. It would be useful to have, at the beginning of the results, an explanation of what the HR and RD values actually mean. For example, the “considerable absolute rate differences ($RD \geq 0.49$)” presumably means there are 0.49 more cases of the condition per 1,000 person-years in the eczema group. How was the considerable absolute rate differences ($RD \geq 0.49$) decided? It seems quite small. Can you present the absolute rates for each outcome by group graphically?
- The discussion section on benchmarking against previous studies was a good idea, apart from the fact that they were all from the same CPRD database which may be susceptible to the same potential biases. It would have been nice to have seen benchmarking against other cohort studies, of which there are now many.
- There seems to be too much attention paid to whether the findings are in line with the AAD guidelines as if they were the last word on atopic dermatitis care globally. We appreciate that they were used to identify putative outcomes, but to structure much of the discussion on whether the findings were suggested in the AAD guidelines or not seems a bit over the top.
- In limitations it would be useful to provide details on the validation so the reader has an idea of how accurate the diagnosis code is (e.g. 86% PPV in survey of 200 patients).
- What is the clinical utility of these associations? – guidelines are mentioned, but what will the guidelines say? How does “being aware” of a small increased risk of over 50 conditions for example alter patient management? Does the magnitude of these associations warrant a lower threshold for additional investigations? How would such a strategy risk over-diagnosis of incidental but asymptomatic GORD for example? (in which a 1.1 risk difference was observed in Fig 2)
- If there were specific outcomes that are amenable to primary or secondary prevention (eg shingles vaccine for an increased risk of zoster) then some form of action would be possible.
- It is not clear which, if any, of the associations might be treatment related. Fractures for example might be related to increased use of potent topical corticosteroids (or inhaled corticosteroids for associated rhinitis/asthma)
- Reproducibility. Data access issues limit the reproducibility, understandably. However, it would be useful to provide the exact code use (rather than dummy examples) so that someone who does have access to the data can run it (especially as its Make-like) to ensure that the results could be reproduced.

Reviewer #2

(Remarks to the Author)

Reviewer #3

(Remarks to the Author)

This is a well performed study of a wide variety of comorbidities in AD.

Is there a specific reason you decided not to do a PheWAS? This approach would have given you a hypothesis free approach. In my opinion, you have over interpreted some of your findings such as the irritable bowel, reflux, oesophagitis, etc which could have been explained by a surveillance bias given the fact that you studied a cohort that visited the GP. Is there a way in which you can assess the impact of this potential bias and if not, I would more carefully interpret your findings (eg, the changes that oesophagitis is an eosinophilic oesophagitis is small)

Could you provide an estimate of the residual confounding in your study? Important confounders such as life style factors and drug exposure have not been taken into account and might explain some of your findings.

Although you have applied a specific statistical method to address Berkson bias, but could you elaborate on how this might have affected your results?

Version 1:

Reviewer comments:

Reviewer #1

(Remarks to the Author)

I am impressed by the way that the authors have responded to my reviewer comments. They have considered them carefully and undertaken additional analyses and new presentation of the data that adds clarity and credibility. As a result the article is now better structured and much more balanced with more cautious interpretation in the right places in the discussion section. The abstract is also a lot more logical and helpful. Pity you were not able to look at the specificity of findings eg by comparison with psoriasis, as I am still sceptical that some of the associations are an epiphenomenon of the large data base (with some degree of surveillance bias and residual confounding), but I think you have now "done enough" to encourage others to pursue such an approach.

Reviewer #2

(Remarks to the Author)

Reviewer #3

(Remarks to the Author)

REVIEWER COMMENTS

Reviewer #1 (Reviewers: Hywel C. Williams and Lloyd Steele)

COMMENT 1.1: General comments

This submission represents a large body of work produced by international experts in the field of routine healthcare records.

Importantly, the team also have content expertise in atopic eczema.

We liked the general epidemiological approach of investigating multiple outcomes in the same study as an efficient and cost-effective way of addressing multiple questions.

Much thought has been given to sensitivity analyses.

The work is also well reported.

The study will be a useful benchmark for other national databases to try and replicate

RESPONSE 1.1: We thank the reviewers for their thorough review and the very useful comments that have helped us substantially improve the manuscript.

Scientific credibility

COMMENT 1.2: There is a hint in the writing style of the manuscript that the many studies conducted to date have been a waste of time “The existing evidence base, encompassing hundreds of studies employing heterogeneous approaches to study design, analysis and data management, hinders comparison, and has been slow and expensive to build”. Sometimes, consistency from a range of study designs is a strength. Sometimes doing things slowly taking into account previous research can also be a strength.

RESPONSE 1.2: We agree, and have therefore changed the sentence to reflect this.

CHANGES 1.2:

- In the Abstract > Background, we replaced “*The existing evidence base, encompassing studies employing heterogeneous approaches to study design, analysis and data management, hinders comparison, and has been slow and expensive to build.*” with “*Atopic eczema may be related to multiple subsequent adverse health outcomes, however, evidence from large population-based studies across outcomes, and comparisons between outcomes, are needed.*”

COMMENT 1.3: Abstract: The chronology of the inception disease (eczema) and subsequent outcomes is unclear in the abstract. Asthma and allergic rhinitis (the most obvious associated diseases that tend to develop later in the so called allergic march) are not mentioned in the abstract.

RESPONSE 1.3: We have clarified the chronology of disease and restructured the overview of results in the abstract.

CHANGES 1.3:

- Abstract > Results now reads: “*Eczema was associated with the subsequent diagnosis of outcomes with adjusted HRs (99% confidence intervals) of up to 4.02(3.95-4.10) for food allergy (RD per 1,000 person-years of 1.5). Besides strong*

associations with atopic and allergic conditions (e.g., asthma 1.87[1.39-1.82], RD5.4; allergic rhinitis 1.92[1.90-1.93], RD5.3) and skin infections (e.g., molluscum contagiosum 1.81[1.64-1.96], RD1.8), the strongest associations were with Hodgkin's lymphoma (1.85[1.66-2.06], RD0.02), Alopecia Areata (1.77[1.71-1.83], RD0.2), Crohn's disease (1.62[1.54-1.69], RD0.1), Urticaria (1.58[1.57-1.60], RD1.9), Coeliac disease (1.42[1.37-1.47], RD0.1), Ulcerative colitis (1.40[1.34-1.46], RD0.1), Autoimmune liver disease (1.32[1.21-1.43], RD0.01), and Irritable bowel syndrome (1.31[1.29-1.32], RD0.7). Sensitivity analyses revealed the impact of consultation bias or choice of cohort age cut-off on findings. Severity analyses showed comparatively large HRs in severe eczema for some liver, gastrointestinal and cardiovascular conditions, osteoporosis, and fractures. Most cancers and neurological conditions were not associated with eczema."

COMMENT 1.4: Severity. Is there any evidence for the approach for severity? A single prescription of 30g Betnovate would presumably lead to a diagnosis of "moderate-severe" eczema, but with many adults included this could be for hand eczema? In the validation approach above, we see that none of the 200 patients had severe disease and only 25% had symptoms of eczema in the past year (which seems like a very mild cohort).

RESPONSE 1.4: We agree that there were concerns about the moderate-to-severe cohort potentially not representing a cohort of people with moderate-to-severe disease. We have therefore changed our approach to severity, and now report estimates from a three-level severity scale, based on prescriptions and secondary care admissions. We are now more confident that the cohort of individuals with severe eczema have more severe eczema, since in the UK, only those with the most severe eczema are seen in secondary care.(1) The new approach to severity also has the advantage of mirroring approaches used by previous studies in CPRD, allowing us to benchmark severity results as we did for the main results (**Supplementary Figure 2**). In our updated severity analyses, reassuringly we again see that our results are similar to those from previous studies. A limitation of the approach is that individuals may have eczema recorded in secondary care when they are admitted to hospital due to another condition. To address this concern, we additionally adjusted models for time-updated secondary care admissions, which we present together with comorbidity-adjusted models in **Supplementary Table 2**. Additional limitations are that severity analyses are now limited to individuals who were eligible for linkage to secondary care records, and the unavailability of outpatient records for this study.

We acknowledge that despite improvements, our approach to measuring eczema severity still has its drawbacks (as it does in most EHR studies due to inherent limitations related to the reasons for data capture). However, use of population-based routinely-collected data to answer these questions remains an advantage over data solely from secondary or tertiary care referral contexts. We therefore encourage cautious interpretation of the results of our analyses by eczema severity. Our results should be used as an additional tool to judge the strength of evidence, rather than representing precise estimates of the increase in risk in people with mild, moderate or severe eczema. We have added clarification in the discussion section.

CHANGES 1.4:

- In Methods, Study population, we added: "For secondary analyses, eczema severity was defined as mild, moderate, or severe as a time-updated variable. People with

eczema were assumed to have mild disease in the absence of any evidence for moderate or severe disease. When they received prescriptions for potent topical corticosteroids or topical calcineurin inhibitors (after meeting the eczema definition), they were considered as having moderate eczema. When they had eczema recorded in secondary care, received prescriptions for systemic immunosuppressants (azathioprine, cyclosporine, methotrexate, or mycophenolate mofetil), or had phototherapy recorded, they were considered as having severe eczema. Individuals' assigned severity could progress from mild to moderate to severe, but not revert from moderate or severe to mild, or from severe to moderate; hence this variable denoted whether a person had ever-experienced moderate or severe eczema, which is an approach used in several previous studies."

- Updated Figure 3 to include the new 3-scale severity results.
- Updated Supplementary Table 2 to include the new 3-scale severity results, with both comorbidity adjusted and additionally time-updated hospital admission adjusted results.
- Added Supplementary Figure 2 showing benchmarking of severity analysis.
- In Discussion > Limitations, we added: *"We were limited to defining severity based on prescriptions and hospital admissions, which can only approximate severity. We therefore encourage cautious interpretation of effect estimates, and that results should be used as an additional tool to judge the strength of evidence, rather than representing precise estimates of the increase in risk in people with moderate or severe eczema."*
- In Results > Benchmarking against previous studies, we added: *"We also found that our results from analyses by eczema severity were generally similar to results from these studies (Supplementary Figure 2)."*
- In Discussion > Interpretation, we added: *"Future research may aim to investigate mechanisms through which eczema, and especially more severe eczema, may be associated with outcomes [...]"*

COMMENT 1.5: Distribution of age at index date and follow-up. The median age of 24 for index case in eczema seemed high. Can you show the distribution of this? We also found it difficult to link the long-term outcomes (e.g. dementia) with the relative short follow-up period, so it would also be useful to see the distribution of the follow-up time.

RESPONSE 1.5: The median age of 24 does not reflect the median age of onset of eczema in the population under study. Rather, this is the median age at which individuals (that have been marked as research eligible by CPRD) first meet the eczema definition (of having at least one diagnosis code for eczema and at least 2 prescriptions for eczema on different days) while they are contributing data to CPRD. Individuals in CPRD rarely have data available for their entire life from birth; rather, an individual's follow-up time reflects the time they were registered at a GP practice, which can vary.

We now report histograms of both age at index date and follow-up time for exposed individuals (Supplementary Figures 3 and 4). The histogram of age at index date reveals a jump at 18 years of age, which may be explained by people changing their GP practice around this time. Supplementary Figure 3 shows that a substantial number of individuals are aged 60 or older at their indexdate, and Supplementary Figure 4 shows that the follow-up time for many individuals extends to more than 20 years. Therefore, our study is suitable to

assess associations even with long-term outcomes typically occurring in older age, such as dementia.

CHANGES 1.5:

- Added Supplementary Figure 3: Histograms of age at index date.
- Added Supplementary Figure 4: Histograms of follow-up time.
- In Results > Descriptive Statistics, we added: “[...] (see histograms in Supplementary Figure 3 [age at index date], and Supplementary Figure 4 [follow-up time]).”

COMMENT 1.6: Most of the associations are not novel eg the links between atopic eczema and inflammatory bowel disease. We found 66 citations on this topic including a reasonably well conducted systematic review that confirmed the bidirectional association between atopic eczema and IBD (Lee H et al Bidirectional relationship between atopic dermatitis and inflammatory bowel disease: A systematic review and meta-analysis. J Am Acad Dermatol. 2020;83:1385-1394). The authors should state clearly what the really novel associations discovered in this important piece of work are in the abstract or perhaps state that the work is more confirmatory than novel in terms of the associations found. What is novel is the comprehensiveness and precision and representativeness of the associations and some degree of dose response in relation to severity based on associated treatments. The presentation of absolute risks is also valuable for estimating the magnitude of the health issues to individuals and public health planners.

RESPONSE 1.6: We agree that we previously relied too heavily on the AAD guidelines to determine which associations we should consider as novel, which also relates to Comment 1.18. Following reviewer comments, we have changed our approach to discussing the results.

CHANGES 1.6:

- We have rewritten the abstract to include an overview of all outcomes, rather than focusing on those that were missing from the AAD guidelines (as is covered in response to COMMENT 1.3)
- We have changed the structure of the discussion section; we replaced the sections on “*Associations considered in the AAD guidelines*” and “*Associations not considered in the AAD guidelines*” with a section on “*Discussion of findings by category*”, which now also incorporates new findings from the new sensitivity and severity analyses.
- The “*Summary of the most relevant findings*” section now reads: “*Besides confirming associations with atopic conditions, immune-mediated skin conditions and skin infections, we found strong associations with Hodgkin’s lymphoma, Crohn’s disease, coeliac disease, ulcerative colitis, and autoimmune liver disease, albeit with relatively small absolute rate differences for these outcomes. More common, but less strongly associated, outcomes included irritable bowel syndrome, oesophagitis, gastro oesophageal reflux disease, thromboembolic disease, obesity, gastritis and duodenitis and peripheral neuropathies. Our severity analyses also suggest that some outcomes may be primarily associated with severe eczema, and not all eczema, for example cardiovascular outcomes, osteoporosis and fractures.*”
- The “*Discussion of findings by category* section” now reads: “*The largest associations were found with atopic and allergic conditions, urticaria and alopecia areata, which is already well known from clinical practice, and recognised in guidelines on awareness of eczema comorbidities.(2) We found evidence of a link*

with skin infection, which is also well known clinically, staphylococcus infection being a diagnostic criterion for eczema.(3) We also found an association with COPD, however the increased HR in the <18 cohort suggests that there may be overlap with asthma recordings, as COPD usually occurs in older age.

We found evidence for associations with autoimmune liver disease and liver fibrosis/sclerosis/cirrhosis, albeit with small rate differences, and fatty liver, with a larger, but still relatively small, rate difference. Sensitivity analyses suggest that at least some of the effect seen may be explained by consultation bias, especially for fatty liver. We saw dose-response relationships with eczema severity and very large HRs for severe eczema, suggesting that most of the increased risk may be in those with severe eczema. We found little existing evidence on these associations, so it is likely there was little awareness of these potential links, however given the relatively small rate differences these outcomes may be less important to consider in screening and prevention contexts.

We found strong evidence for associations with inflammatory bowel diseases, that held up in sensitivity analyses. We also saw risk increasing with more severe eczema, with some of the largest effects for severe eczema seen across all outcomes. A recent study from UK population data showed similar results,(4) and other studies have similarly concluded that a, possibly bidirectional, association exists.(5) The comparison with other outcomes in our study suggests that inflammatory bowel diseases may be some of the most relevant to consider for future research, however, small rate differences may suggest less relevance in informing screening or prevention measures in people with eczema.

We also found evidence for diseases of the oesophagus, albeit with less clear dose-response relationships with eczema severity. Some previous population-based studies have shown similar results, e.g., for gastro-oesophageal reflux,(6) however, findings may be partially explained by an increased risk of developing eosinophilic oesophagitis, for which awareness is increasing but may still be misdiagnosed.(7) Of other digestive system conditions studied, the evidence of association was strongest for coeliac disease, which has previously been studied together with other autoimmune conditions.(8) For Irritable bowel syndrome, and gastritis and duodenitis, the evidence of association was also relatively strong considering strength of associations, and results from sensitivity and severity analyses.

Our results suggest small relative, but potentially considerable absolute, increased risks for depression and anxiety; for alcohol abuse and cigarette smoking the evidence was weaker.

Our findings suggest uncertain evidence and/or weak associations with autism and attention deficit hyperactivity disorder (ADHD) in line with existing guidelines.(2) Our findings for autism in particular should be interpreted with caution as results from analyses where the 40+ cohort were used showed a large increase in the hazard ratio, which is unexpected, given autism is usually diagnosed in childhood, and it is unlikely people with eczema would have higher rates of autism in adulthood.

We found a somewhat increased risk of thromboembolic (e.g., deep vein thrombosis, phlebitis) and peripheral artery disease, with weaker evidence for heart failure, coronary artery disease and hypertension, and only very weak, or for a very small increased risk, for stroke and myocardial infarction. For some conditions, e.g., hypertension, increased risk may be almost entirely explained by consultation bias. We saw relatively large HRs for severe eczema for cardiovascular outcomes, as was the case in a previous population-based study.(9)

While we saw an association with obesity, we did not see a dose-response

relationship with eczema severity, and again saw that consultation bias may be an important explanatory factor; this was similar for metabolic syndrome albeit with few events occurring in our study. While for diabetes we saw risk increasing with more severe eczema, in the main analysis the risk was relatively small, possibly explained by consultation bias, and we saw a null result when using the <18 cohort. This may suggest that eczema has little or no effect on diabetes occurring in younger age, but may still have an effect on diabetes occurring in older age.

In existing guidelines, the association with osteoporosis was graded as being of high certainty,(2) based on three studies,(10,11) one population-based matched cohort study from Taiwan(12) showing HRs of more than 4 (as compared to our HR of 1.18 [1.16-1.20]). While in our study there was evidence of only small increases of risk for osteoporosis and fractures (compared to other outcomes), we found relatively large HRs for severe eczema, as was the case in a previous study,(13) suggesting risk may potentially only be increased in those with severe eczema.

We found a relatively large relative and absolute effects for peripheral neuropathies, about half of the records that made up this outcome being for sciatica. There was also some evidence for an association with migraine, a recent study showing similar effect sizes (HR from fully adjusted model 1.2 [1.2 – 1.26]) to ours (aHR 1.18 [1.17-1.19]).(14) However, sensitivity analyses suggest these associations may be explained in large part by consultation bias.

Our findings are consistent with those from a previous study that showed no evidence for association with solid organ cancers but associations with lymphomas.(15) The larger sample size of our study allowed more precisely estimation of the association with Hodgkin's lymphoma, which has one of the largest effect estimates of any outcomes, but a low absolute difference."

- *The "Conclusion" section now reads: "We give a comprehensive overview of adverse health outcomes associated with eczema, for each of the 71 outcomes providing evidence from a large and representative database, including from sensitivity analyses and analyses by eczema severity. The cross-outcome approach offers additional benefits of comparability between results, reduced investigator biases and efficiency of evidence generation. Findings can be used to inform guidelines and clinical practice, and as a baseline for more detailed research on individual outcomes."*

COMMENT 1.7: Given that eczema is such a common disease of childhood, it is odd that a children's cohort has not been examined specifically (instead of being lumped into the all ages group), especially given increased confidence in the diagnosis of atopic eczema at this age (as opposed to irritant contact dermatitis, allergic contact dermatitis, varicose eczema, asteatotic eczema etc that are much more common as a differential diagnosis in adults that might be diagnosed in primary care as just "eczema" and treated with the same topical treatments as atopic eczema)

RESPONSE 1.7: We agree that adding a cohort where eczema is diagnosed in childhood offers additional value as a sensitivity analysis, in representing a cohort of people with more definite eczema and in representing a younger cohort. We now present findings from the cohort of people with eczema diagnosed in childhood (" <18 cohort"). In summary, sensitivity analyses with the <18 cohort showed larger aHRs for food allergy, allergic rhinitis, asthma and Crohn's disease than the respective main analyses. For most other outcomes we see an attenuation of aHRs as compared to the main analysis. Finally, for outcomes where we considered the 40+ cohorts the main cohort, and for most cancer and liver outcomes, and

metabolic syndrome, we saw that the number of outcomes was too small to make inferences. In Figure 3, we therefore omit results from all analyses with the <18 cohort where the number of outcome events in people with eczema was below 1,000.

Follow-up in the <18 cohort is not limited to individuals' 18th birthday, and the cohort also includes a small proportion of individuals (~13%) who are first considered to have eczema before their 18th birthday but only start follow-up after their 18th birthday due to not meeting CPRD eligibility criteria.

CHANGES 1.7:

- Updated Figure 3 to include results from the <18 cohort.
- Updated Supplementary Table 3 to include results from the <18 cohort.
- Updated Methods > Study population (3rd paragraph): *“For sensitivity analyses, we created two additional cohorts. Firstly, [...]. Secondly, a subset of the any age cohort of individuals that met the eczema definition before their 18th birthday (<18 cohort).”*
- In Results > Associations between eczema and adverse health outcomes, we added: *“Results from the <18 cohort varied; for atopic and allergic outcomes HRs were larger, while for several other outcomes HRs were attenuated as compared to their respective main results.”*

COMMENT 1.8: Specificity: It is not clear how specific these findings are to atopic eczema – perhaps similar findings would emerge for other chronic inflammatory or infectious conditions (eg psoriasis, rheumatoid, periodontal disease) when examined in such a way in a massive database. Comparing atopic eczema to the general population is a good start, but we would have liked to have seen some “active” controls too to minimise the possibility of generation of spurious associations.

RESPONSE 1.8: We agree that a comparison with another inflammatory conditions (e.g., psoriasis) would be of great interest and could provide key evidence on the nature of (systemic) inflammation of eczema. While the already large scope of this study, capacity constraints, and the scope of the current approval did not allow us to pursue this for this manuscript, it should be perused in future research. We have added a sentence jointly addressing this comment and COMMENT 1.9 to the discussion section.

CHANGES 1.8: In Discussion > Interpretation, we added: *“[...] the role of eczema as a systemic disorder associated with systemic inflammation, and the extent to which outcomes are shared with other chronic inflammatory conditions (e.g., psoriasis).”*

COMMENT 1.9: Perhaps the many outcomes in this study are shared by the family of conditions characterised by T helper cell 2 inflammatory responses – such a notion might be explored in some follow-on research.

RESPONSE 1.9: We have added a sentence jointly addressing COMMENT 1.8 and this comment to the discussion section.

CHANGES 1.9: See CHANGES 1.8.

COMMENT 1.10: Fig 2 shows that all 71 conditions are significantly increased with atopic eczema (with the exception of prostate cancer and Parkinson’s disease which

did not quite reach statistical significance). We appreciate this is probably because the team only studied those diseases that were rumoured in various guidelines and studies to be associated with eczema (so those rumours were right after all!), but we are left wondering whether the pattern in Fig 2 is an epiphenomenon of large routine health data research that examines just one disease? We would have found the specificity of the findings more convincing if the risk estimates had been more scattered and some were negative (eg previous studies have suggested a lower risk of brain cancer). We did not see any evidence of negative controls as hinted in the discussion as almost everything turned out to be positive.

RESPONSE 1.10: While it is true that most conditions studied were “significantly” associated with eczema, we feel like the notion of a significance cut-off (i.e., the confidence intervals not overlapping the null) is not very useful in the context of very large sample size and residual biases. We agree that this may indeed be an epiphenomenon of large routine health data research, and that our study reveals this issue more clearly than would studies on individual outcomes (which may have produced “significant” results, regardless of which outcome was studied). In our interpretation of results, whether or not a result is significant plays only a small role; rather, the strength of association, including the size of the absolute risk estimates, together with findings from sensitivity and severity analyses are considered.

In addition, as described below in the response to COMMENT 1.12, we have now added sensitivity analyses excluding “non-consulting” individuals. Here, in addition to Parkinson’s disease and prostate cancer, several additional outcomes do not cross the significance threshold, including oesophageal varices, cholecystitis, pancreatitis, appendicitis, peritonitis, metabolic syndrome, pelvis fracture, multiple sclerosis, vascular dementia, Alzheimer’s dementia, myeloma, melanoma, CNS cancers, lung cancer and breast cancer. Previously, we had suggested that there was no strong evidence for an association with any of these outcomes, as effect estimates were close to the null, which is further justified by these new results.

CHANGES 1.10: In Discussion > Interpretation, we added: “Our results can be used to judge the plausibility and strength of links between eczema and the subsequent development of adverse health outcomes. Rather than relying solely on a statistical significance threshold, which due to high power may be met for unimportant or small effects, the strength of association together with findings from sensitivity and severity analyses should be considered. Absolute measures of effect allow judging the potential public health relevance. Uniquely, across all analyses our study offers a comparison between outcomes. Whether associations are causal, implying effective diagnosis and treatment for eczema could prevent the development of these comorbidities, is not possible to determine from this study alone. [...]”

COMMENT 1.11: Significant results: It can be difficult to ascertain what is a positive result and negative result. Some examples are: Diabetes. The discussion reports “Our findings would support stating that eczema may not be associated with diabetes”. The results for diabetes show aHR 1.12 [99% interval 1.11-1.14] and RD 0.43 (which is a larger RD than reported for several diseases classed as significant). Myeloma. Eczema was weakly associated with myeloma (aHR 1.11 [1.04-1.19]; RD 0.01), We found evidence for associations with autoimmune liver disease and liver fibrosis/sclerosis/cirrhosis, albeit with small rate differences (results were aHR 1.35 [1.24-1.47]; RD 0.02) and (aHR 1.27 [1.22-1.33]; RD 0.09), respectively). Lymphoma is reported as a less common but more strongly associated outcomes, but the RD is

only 0.02-0.05. It can be quite unclear where the line is drawn for diabetes not being associated but autoimmune liver disease being reported as a new significant association.

RESPONSE 1.11: We agree that differentiating between negative results and weakly associated positive results is not always possible. In the examples given in the comment, we judged whether an outcome was associated primarily on the estimated strength of the association (magnitude of the HR), and not the potential public health importance of such an association (magnitude of the RD); which is why we would argue that it is reasonable to be more confident in associations with autoimmune liver disease and liver fibrosis/sclerosis/cirrhosis (with HRs around 1.3), than for diabetes (with a HR of 1.12).

However, for all outcomes our study provides additional evidence next to the HR. For the example of diabetes, we now have three additional pieces of evidence. Firstly, the analysis excluding “non-consulters” showed attenuated effect estimate (to 1.07 [1.05-1.08]), as was the case for almost all other outcomes. Secondly, the analysis using the <18 cohort showed attenuated effect estimates (with 99% CIs crossing the null), which was not the case for most other outcomes. This may suggest that eczema has little or no effect on diabetes occurring in younger age, but may still have an effect on diabetes occurring in older age. Thirdly, in the new severity analysis, we see a dose-response relationship between eczema severity and the risk of diabetes (mild: 1.03 [1.01-1.05]; moderate: 1.11 [1.09-1.12]; severe: 1.18 [1.11-1.27]). However, this dose-response relationship is small compared to dose-response relationships seen for other outcomes with similar HRs in the main analysis, e.g., cardiovascular outcomes.

While we describe all results for each outcome in Supplementary Notes 1, which is now updated to include new results from this revision, we are limited by the manuscript word count for the inclusion of a full discussion for each outcome. However, we have restructured the discussion section, as described in RESPONSE 1.6, which we feel addresses some of your concerns.

CHANGES 1.11: See CHANGES 1.6.

COMMENT 1.12: Surveillance bias due to increased healthcare contact from the index condition is not mentioned at all in the submission.

RESPONSE 1.12: Thank you for this important suggestion, which was also brought up in COMMENT 3.3. To address this shortcoming of the study, we have now performed an additional analysis excluding “non-consulters”. Here, individuals who do not have any record for 9N11.00 (Seen in GP's surgery), 22A..00 (Body weight), 4....00 (Laboratory test), or 246..00 (O/E - blood pressure reading) in the year prior to index date are excluded. These records were selected as they are unlikely to denote any specific disease, and jointly they occur approximately 750,000,000 times in CPRD Aurum (according to the CPRD Aurum code browser). While an analysis excluding individuals without any record in the year prior to index date may have been preferable, we were limited to using the four specified codes due to data extraction restrictions.

In summary, we believe this sensitivity analysis suggests that at least some of the effects seen in the main analysis could be explained by consultation bias. We had previously applied a cautious approach to judging causality, and all strongly associated outcomes remain strongly associated in analyses excluding non-consulters. Therefore, there is little overall change in the discussion of which outcomes are associated. However, we believe

that our study now offers stronger evidence for outcomes that remain strongly associated in this new sensitivity analysis, and we thank the Reviewers for this important comment.

CHANGES 1.12:

- Added hazard ratios from analysis excluding non-consulters to Figure 3.
- Added hazard ratios from analysis excluding non-consulters to Supplementary Table 1.
- Added to discussion as described in CHANGES 1.6.
- In Methods > Study Population, we added: *"In an additional sensitivity analysis performed to address consultation bias, using each outcome's respective main cohort, individuals who did not have any of four common records (9N11.00 - Seen in GP's surgery, 22A..00 - Body weight), 4....00 - Laboratory test, 246..00 - O/E - blood pressure reading) in the year prior to index date were excluded."*

COMMENT 1.13: Multiple testing: The authors report "To account for multiple testing, we reported wider 99% instead of usual 95% confidence intervals", but this is not quite the same as correcting for multiple testing. They report a Bonferonni corrected p-value, but every outcome in the minimally-adjusted model are below this p-value (that is, $p < 0.0007$) and in 65/71 of the comorbidity-adjusted model. While the authors rightly do not focus on significance here, multiple testing appears to be an issue.

RESPONSE 1.13: There is no universally agreed approach to handle multiple testing, and in this manuscript we chose two approaches: firstly using wider confidence intervals compared to what is commonly done in studies using electronic health records, and secondly, by reporting a Bonferroni corrected p-value. Due to the large sample size, we still see very small p-values and, as the reviewer notes, most p-values are still "significant" even under a Bonferroni threshold. This is not necessarily a sign that multiple testing is an issue, as the aim of these corrections is not to remove some set number of significant findings but to make sure that those hypothesis tests can be judged against a more stringent threshold. Indeed, Bonferroni corrections are often criticised for being overly conservative.⁽¹⁶⁾ We've discussed the impact of multiple testing, but note that this is also an issue in the usual conduct of multiple separate studies on different outcomes in the same database. We feel that, while the usually occurring 'community-wide multiple testing' is usually invisible and un-addressed, the number of tests performed in our study is transparent, offering some opportunity to account for multiple testing. This is captured in our Discussion > Strengths section as follows: *"Firstly, in epidemiological research, hundreds of tests across multiple studies are often performed using the same data source. However, multiple testing is rarely considered since these tests are done across many different studies. In our study it was straightforward to include adjustments for multiple testing (although this was less important to consider in our study given the large sample size supplied high power to test multiple outcomes)."*

CHANGES 1.13: No changes.

Interpretation and clinical utility of the data

COMMENT 1.14: Biological plausibility is often a retrospective exercise, and sometimes associations that do not appear plausible are found to have some plausibility at a later time when more is known about the disease. But some

discussion on biological plausibility for the outcomes or groups of outcomes would be nice.

RESPONSE 1.14: We agree that a discussion of biological plausibility is important, however given the limited word count of the manuscript we believe such discussions are better suited for other formats, e.g., a commentary on this article or a systematic review, which then also can take other literature into account.

CHANGES 1.14: In the last paragraph of Discussion > Limitations, we added: “[Individual studies] [...] may benefit from more detailed application of expert knowledge, including reviews of the existing literature and discussion of biological plausibility, to each exposure-outcome relationship.”

COMMENT 1.15: We like the idea of using multiple cohorts within a national database of routine health data to sort of verify the consistency of findings, but it is not quite the same as replication in another routine health data set that has been set up in a different way. So we hope that this work will be replicated by others with the skills to interrogate other large routine health data sources such as Taiwan.

RESPONSE 1.15: We wholeheartedly agree that this type of work should be replicated in other databases, such as the Taiwan National Health Insurance Research dataset and have emphasised this in the discussion.

CHANGES 1.15: In Discussion > Interpretation, we added: “*Replicating this work in other large population-based data sources will further strengthen the evidence.*”

COMMENT 1.16: Context to values. It would be useful to have, at the beginning of the results, an explanation of what the HR and RD values actually mean. For example, the “considerable absolute rate differences (RD ≥ 0.49)” presumably means there are 0.49 more cases of the condition per 1,000 person-years in the eczema group. How was the considerable absolute rate differences (RD ≥ 0.49) decided? It seems quite small. Can you present the absolute rates for each outcome by group graphically?

RESPONSE 1.16: We have added clarification of the hazard ratio and rate difference. The rate difference cut-off was chosen after considering rate differences of outcomes with stronger evidence (i.e., with largest main aHRs that held up in sensitivity analyses and signals of dose response from severity analyses). Several of these outcomes had rate differences just above 0.5 (e.g., 0.67 for irritable bowel syndrome, 0.51 for thromboembolic diseases, 0.6 for gastritis and duodenitis), while several other strong-evidence outcomes had rate differences ranging from 0.01 to 0.18 (e.g., 0.02 for Hodgkin lymphoma, 0.18 for Alopecia Areata, 0.09 for Crohn’s disease, 0.10 for Coeliac disease). We recognise this is an arbitrary cut-off, however, one that fulfils a purpose in giving readers context needed to judge potential implications for clinical practice and public health. We also acknowledge that other cut-offs could be used to separate outcomes into groups, e.g., rarer vs more common outcomes (e.g., based on absolute rates). Given the relationship between absolute rates and rate differences, there would likely be little difference in grouping of outcomes compared to the current grouping based on the rate difference cut-off. We have removed mention of a RD cut-off in the Abstract, and now only group outcomes into those with “relatively small rate differences” and “more common” outcomes in the discussion section.

CHANGES 1.16:

- In Results > Associations between eczema and adverse outcomes we changed the first sentence to: *“For all outcomes, comorbidity-adjusted hazard ratios (i.e., the relative increase in hazard in the exposed) with 99% confidence intervals, and estimated rate differences (RD) (i.e., the number of additional outcomes experienced by the exposed) per 1,000 person-years [...]”*
- The Discussion > *“Summary of most relevant findings”* section now reads: *“Besides confirming associations with atopic conditions, immune-mediated skin conditions and skin infections, we found strong associations with Hodgkin’s lymphoma, Crohn’s disease, coeliac disease, ulcerative colitis, and autoimmune liver disease, albeit with relatively small absolute rate differences for these outcomes. More common, but less strongly associated, outcomes included irritable bowel syndrome, oesophagitis, gastro oesophageal reflux disease, thromboembolic disease, obesity, gastritis and duodenitis and peripheral neuropathies. Our severity analyses also suggest that some outcomes may be primarily associated with severe eczema, and not all eczema, for example cardiovascular outcomes, osteoporosis and fractures.”*

COMMENT 1.17: The discussion section on benchmarking against previous studies was a good idea, apart from the fact that they were all from the same CPRD database which may be susceptible to the same potential biases. It would have been nice to have seen benchmarking against other cohort studies, of which there are now many.

RESPONSE 1.17: The studies used to benchmark our results were intentionally chosen to be from the same data source and using a similar study design. The benchmarking was intended to, firstly, provide reassurance of correctly implemented study design, and secondly, evaluate whether the effects estimated using an outcome-wide confounding adjustment strategy were similar to those estimated using bespoke confounder-selection strategies. While a systematic review and/or meta-analysis comparing results from this study to results from other studies would certainly be of interest, it is beyond the scope of this work.

CHANGES 1.17: No changes.

COMMENT 1.18: There seems to be too much attention paid to whether the findings are in line with the AAD guidelines as if they were the last word on atopic dermatitis care globally. We appreciate that they were used to identify putative outcomes, but to structure much of the discussion on whether the findings were suggested in the AAD guidelines or not seems a bit over the top.

RESPONSE 1.18: As described in RESPONSE 1.6 (on novelty of associations), we have shifted the focus in the discussion towards novelty of the comprehensiveness, precision and comparability this study provides and away from the novelty of individual associations (which was based on the AAD guidelines). As also mentioned in COMMENT 1.17, a comparison with other studies in the form of a systematic review and/or meta-analysis would be of interest but is beyond the scope of this study.

CHANGES 1.18: Relevant changes described in CHANGES 1.6.

COMMENT 1.19: In limitations it would be useful to provide details on the validation so the reader has an idea of how accurate the diagnosis code is (e.g. 86% PPV in survey of 200 patients).

RESPONSE 1.19: We have added a sentence describing a validation study our eczema definition is based on.

CHANGES 1.19:

- In Methods > Study Population, we added: *“An analogous algorithm has been previously validated in UK primary care data and was found to have a positive predictive value of 86%.”*

COMMENT 1.20: What is the clinical utility of these associations? – guidelines are mentioned, but what will the guidelines say? How does “being aware” of a small increased risk of over 50 conditions for example alter patient management? Does the magnitude of these associations warrant a lower threshold for additional investigations? How would such a strategy risk over-diagnosis of incidental but asymptomatic GORD for example? (in which a 1.1 risk difference was observed in Fig 2)

RESPONSE 1.20: Part of the utility of this study is showing not just which outcomes are statistically associated, but also showing which outcomes are associated with comparatively larger relative and absolute increases in risk, increased risk in more severe eczema, and consistent interpretations across a range of sensitivity analyses. This is why, in the discussion, we do not suggest that clinicians or guideline authors need to be aware of small increases of risk for all different ‘significantly associated’ conditions. Rather, we suggest that this study provides evidence that, in conjunction with evidence from other studies, can be used to inform guidelines.

For example, we find the current guidance on non-atopic comorbidities in the UK’s NICE clinical knowledge summaries lacking in detail (“Other non-atopic comorbidities associated with atopic eczema include allergic contact dermatitis, obesity and cardiovascular disease”). A more useful statement could, rather than applying a binary (associated/not associated) approach, draw comparisons between outcomes in terms of strength of evidence, magnitude of associations, availability of additional evidence (e.g., by severity) to highlight the most important associations. Another obvious place for guideline changes would be the AAD guidelines that are cited in the paper.

CHANGES 1.20:

- We have changed the conclusion; it now reads: *“We give a comprehensive overview of adverse health outcomes associated with eczema, for each of the 71 outcomes providing evidence from a large and representative database, including from sensitivity analyses and analyses by eczema severity. The cross-outcome approach offers additional benefits of comparability between results, reduced investigator biases and efficiency of evidence generation. Findings can be used to inform guidelines and clinical practice, and as a baseline for more detailed research on individual outcomes.”*

COMMENT 1.21: If there were specific outcomes that are amenable to primary or secondary prevention (eg shingles vaccine for an increased risk of zoster) then some form of action would be possible.

RESPONSE 1.21: We agree that vaccine preventable conditions would be an important area of prevention. However, we did not study Herpes zoster or other outcomes amendable to vaccination in this study, thus we didn't include discussion on vaccine preventable outcomes.

CHANGE 1.21: No changes.

COMMENT 1.22: It is not clear which, if any, of the associations might be treatment related. Fractures for example might be related to increased use of potent topical corticosteroids (or inhaled corticosteroids for associated rhinitis/asthma)

RESPONSE 1.22: While our severity analyses may give some indication of treatment-related associations since severity is defined mainly through treatments, mediation through medication is not explicitly examined in this study. Thus, we were not able to disentangle the mechanisms through which eczema may cause adverse outcomes. We believe this is beyond the scope of this study, however, future research may aim to investigate mechanisms (such as treatments for eczema) across multiple outcomes.

CHANGES 1.22:

- In Discussion > Interpretation, we added: *“Future research may aim to investigate mechanisms through which eczema, and especially more severe eczema, may be associated with outcomes, such as sleep deprivation, medications, [...]”*

COMMENT 1.23: Reproducibility. Data access issues limit the reproducibility, understandably. However, it would be useful to provide the exact code use (rather than dummy examples) so that someone who does have access to the data can run it (especially as its Make-like) to ensure that the results could be reproduced.

RESPONSE 1.23: The exact original analysis code is already provided: the “dummy” aspect referred only to the provision of “dummy” data. The dummy data allows interested researchers to run and explore the original code locally even without full data access. We have added clarification to the README files (that can be found on the study's GitHub and Zenodo repositories) that the same code can be used to run analysis on dummy data and the real data, by specifying file paths.

CHANGES 1.23: No changes to the manuscript.

Reviewer #2 (Remarks to the Author):

COMMENT 2.1: I co-reviewed this manuscript with one of the reviewers who provided the listed reports. This is part of the Nature Communications initiative to facilitate training in peer review and to provide appropriate recognition for Early Career Researchers who co-review manuscripts.

RESPONSE 2.1: Thank you.

Reviewer #2 (Remarks on code availability):

COMMENT 2.2: Comment on code included on review (original code to replicate analysis on dataset would be useful).

RESPONSE 2.2: We have addressed this COMMENT in response to COMMENT 1.23.

Reviewer #3 (Remarks to the Author):

COMMENT 3.1: This is a well performed study of a wide variety of comorbidities in AD.

RESPONSE 3.1: Thank you.

COMMENT 3.2: Is there a specific reason you decided not to do a PheWAS? This approach would have given you a hypothesis free approach.

RESPONSE 3.2: There are several reasons why we did not perform a PheWAS: 1. The selection of outcomes is bespoke to the exposure of eczema, chosen to include outcomes that are known to be associated with eczema (e.g., food allergy, asthma), outcomes that have been previously studied in CPRD (to allow benchmarking), and outcomes that were highlighted as priority research areas; these may not have been included in a generic phenotype catalogue. 2. Analyses are adjusted for a range of potentially important demographic and clinical confounders, as well as the status of all other outcomes at index date, which is computationally expensive. Our current analysis pipeline, as it also includes severity and sensitivity analyses, takes weeks to run a high-spec computer (with 128GB RAM), or requires a CPRD-compliant high-memory and high-performance computing cluster to run more quickly. A larger set of outcomes may have required either additional investment in computing resources, or a reduction of the study sample size. This is less of a consideration for a genetic PheWAS as the extent of confounding is more limited for a genetic exposure. 3. The structure of diagnosis recording in CPRD Aurum (in Read and SNOMED terminology) requires bespoke code list creation for each outcome; a hypothesis-free PheWAS would be more suitable to diagnoses encoded in e.g., ICD codes. We agree that a PheWAS (with eczema as the exposure and using a broad catalogue of comorbidities as outcomes) would also offer interesting insights and should be performed in future work.

CHANGES 3.2: No changes.

COMMENT 3.3: In my opinion, you have over interpreted some of your findings such as the irritable bowel, reflux, oesophagitis, etc which could have been explained by a surveillance bias given the fact that you studied a cohort that visited the GP. Is there a way in which you can assess the impact of this potential bias and if not, I would more carefully interpret your findings (eg, the changes that oesophagitis is an eosinophilic oesophagitis is small)

RESPONSE 3.3: We agree that addressing consultation bias is important. As described in COMMENT 1.10 and COMMENT 1.12, we have now implemented an additional sensitivity analysis excluding “non-consulters”. For the specific conditions mentioned (irritable bowel, reflux, oesophagitis), we see an attenuation of effect estimates from analyses excluding non-consulters, suggesting that at least some of the effect may be explained by consultation

behaviour. However, effect estimates remain comparatively large relative to other outcomes, which is reflected in our new discussion section.

CHANGES 3.3:

- Relevant changes concerning the new sensitivity analysis excluding non-consulters described CHANGES 1.12.
- The “*Discussion of findings by category*” on digestive system disorders, other than inflammatory bowel diseases, now reads: “*We also found evidence for diseases of the oesophagus, albeit with less clear dose-response relationships with eczema severity. Some previous population-based studies have shown similar results, e.g., for gastro-oesophageal reflux,(6) however, findings may be partially explained by an increased risk of developing eosinophilic oesophagitis, for which awareness is increasing but may still be misdiagnosed.(7) Of other digestive system conditions studied, the evidence of association was strongest for coeliac disease, which has previously been studied together with other autoimmune conditions.(8) For Irritable bowel syndrome, and gastritis and duodenitis, the evidence of association was also relatively strong considering strength of associations, and results from sensitivity and severity analyses.*”

COMMENT 3.4: Could you provide an estimate of the residual confounding in your study? Important confounders such as life style factors and drug exposure have not been taken into account and might explain some of your findings.

RESPONSE 3.4: We took lifestyle factors and drugs into account, in the form of adjusting for obesity, cigarette smoking and alcohol abuse at baseline, and additionally adjusting for oral glucocorticoid and systemic immunosuppressant use at baseline in a sensitivity analysis. We acknowledge that adjustment for obesity and alcohol abuse do not provide the same granularity as direct adjustment for BMI and alcohol intake, but these measures were not available for this study. This is because, due to the large size of the cohort, we were limited to data on specific pre-defined codes, rather than having access to each individual’s full primary care records. We have added text to the existing paragraph on residual confounding.

CHANGES 3.4:

- In Discussion> Limitations, we added: “*There may be residual confounding through lifestyle factors, not captured in the diagnosis-based smoking, obesity and alcohol abuse definitions.*”

COMMENT 3.5: Although you have applied a specific statistical method to address Berkson bias, but could you elaborate on how this might have affected your results?

RESPONSE 3.5: We usually consider Berkson bias in the context of case-control studies,(17) and we therefore structure our response to this comment around the more generally applicable concept of collider bias.

While the possibility exists that collider bias may be introduced, even when conditioning on a pre-exposure variable, this bias is likely to be small, with some authors suggesting that conditioning on a given pre-exposure variable may generally be the superior choice.(18) Benchmarking against previous CPRD GOLD studies also suggests that major bias through

conditioning on all pre-exposure variables (instead of conditioning on a smaller set of selected covariates) is unlikely, given similar results.

CHANGES 3.5: No changes.

ADDITIONAL RESPONSE: We noticed a minor error in the data management process which led to a small number of individuals' exposure status not being correctly recognised. We have corrected this error which in the main analysis has led to less than 2% difference in individual cohort sizes, with little to no impact on effect estimates. All changes made to the analysis code since the publication of the pre-print can be tracked on the published GitHub repository.

REFERENCES

1. de Lusignan S, Alexander H, Broderick C, Dennis J, McGovern A, Feeney C, et al. Patterns and trends in eczema management in UK primary care (2009–2018): A population-based cohort study. *Clin Exp Allergy*. 2021 Mar;51(3):483–94.
2. Davis DMR, Drucker AM, Alikhan A, Bercovitch L, Cohen DE, Darr JM, et al. AAD Guidelines: awareness of comorbidities associated with atopic dermatitis in adults. *Journal of the American Academy of Dermatology*. 2022 Jan;S0190962222000809.
3. Hanifin JM, Rajka G. Diagnostic Features of Atopic Dermatitis. *Acta Dermato-Venereologica*. 1980 Nov 11;60:44–7.
4. Chiesa Fuxench ZC, Wan J, Wang S, Syed MN, Shin DB, Abuabara K, et al. Risk of Inflammatory Bowel Disease in Patients With Atopic Dermatitis. *JAMA Dermatology*. 2023 Oct 1;159(10):1085–92.
5. Lee H, Lee JH, Koh SJ, Park H. Bidirectional relationship between atopic dermatitis and inflammatory bowel disease: A systematic review and meta-analysis. *Journal of the American Academy of Dermatology*. 2020 Nov 1;83(5):1385–94.
6. Lee SW, Park J, Kim H, Jung YW, Baek YS, Lim Y, et al. Atopic dermatitis and risk of gastroesophageal reflux disease: A nationwide population-based study. *PLoS One*. 2023 Feb 17;18(2):e0281883.
7. Muir A, Falk GW. Eosinophilic Esophagitis: A Review. *JAMA*. 2021 Oct 5;326(13):1310–8.
8. de Lusignan S, Alexander H, Broderick C, Dennis J, McGovern A, Feeney C, et al. Atopic dermatitis and risk of autoimmune conditions: Population-based cohort study. *Journal of Allergy and Clinical Immunology*. 2022 Sep 1;150(3):709–13.
9. Silverwood RJ, Forbes HJ, Abuabara K, Ascott A, Schmidt M, Schmidt SAJ, et al. Severe and predominantly active atopic eczema in adulthood and long term risk of cardiovascular disease: population based cohort study. *BMJ*. 2018 May 23;k1786.
10. Arima K, Gupta S, Gadkari A, Hiragun T, Kono T, Katayama I, et al. Burden of atopic dermatitis in Japanese adults: Analysis of data from the 2013 National Health and Wellness Survey. *The Journal of Dermatology*. 2018;45(4):390–6.

11. Shaheen MS, Silverberg JI. Atopic dermatitis is associated with osteoporosis and osteopenia in older adults. *Journal of the American Academy of Dermatology*. 2019 Feb 1;80(2):550–1.
12. Wu CY, Lu YY, Lu CC, Su YF, Tsai TH, Wu CH. Osteoporosis in adult patients with atopic dermatitis: A nationwide population-based study. Nguyen TV, editor. *PLOS ONE*. 2017 Feb 16;12(2):e0171667.
13. Matthewman J, Mansfield KE, Prieto-Alhambra D, Mulick AR, Smeeth L, Lowe KE, et al. Atopic Eczema–Associated Fracture Risk and Oral Corticosteroids: A Population-Based Cohort Study. *The Journal of Allergy and Clinical Immunology: In Practice* [Internet]. 2021 Sep 24 [cited 2021 Dec 23];0(0). Available from: [https://www.jaci-inpractice.org/article/S2213-2198\(21\)01018-7/fulltext](https://www.jaci-inpractice.org/article/S2213-2198(21)01018-7/fulltext)
14. Lee JH, Kim SH, Lee G na, Han K, Lee JH. Association of atopic dermatitis with new-onset migraine: A nationwide population-based cohort study. *Journal of the European Academy of Dermatology and Venereology*. 2023;37(2):e236–8.
15. Mansfield KE, Schmidt SAJ, Darvalics B, Mulick A, Abuabara K, Wong AYS, et al. Association Between Atopic Eczema and Cancer in England and Denmark. *JAMA Dermatol*. 2020 Oct 1;156(10):1086.
16. VanderWeele TJ, Mathur MB. SOME DESIRABLE PROPERTIES OF THE BONFERRONI CORRECTION: IS THE BONFERRONI CORRECTION REALLY SO BAD? *Am J Epidemiol*. 2019 Mar;188(3):617–8.
17. Snoep JD, Morabia A, Hernández-Díaz S, Hernán MA, Vandenbroucke JP. Commentary: A structural approach to Berkson’s fallacy and a guide to a history of opinions about it. *Int J Epidemiol*. 2014 Apr;43(2):515–21.
18. Ding P, Miratrix LW. To Adjust or Not to Adjust? Sensitivity Analysis of M-Bias and Butterfly-Bias. *Journal of Causal Inference*. 2015 Mar 1;3(1):41–57.